# Neuronal TORC1 modulates longevity via AMPK and cell nonautonomous regulation of mitochondrial dynamics in *C. elegans*

Yue Zhang[1], Anne Lanjuin[1], Suvagata Roy Chowdhury[1], Meeta Mistry[1], Carlos G Silva-García[1], Heather J Weir[1], Chia-Lin Lee[1,2], Caroline C Escoubas[1,3], Emina Tabakovic[1], William B Mair[1]*

[1]Department of Genetics and Complex Diseases, Harvard T. H. Chan School of Public Health, Boston, United States; [2]Department of Biomedical Engineering, National Cheng Kung University, Tainan, Taiwan; [3]Faculty of Medicine, Université Côte d'Azur, CNRS, INSERM, IRCAN, Nice, France

**Abstract** Target of rapamycin complex 1 (TORC1) and AMP-activated protein kinase (AMPK) antagonistically modulate metabolism and aging. However, how they coordinate to determine longevity and if they act via separable mechanisms is unclear. Here, we show that neuronal AMPK is essential for lifespan extension from TORC1 inhibition, and that TORC1 suppression increases lifespan cell non autonomously via distinct mechanisms from global AMPK activation. Lifespan extension by null mutations in genes encoding *raga-1* (RagA) or *rsks-1* (S6K) is fully suppressed by neuronal-specific rescues. Loss of RAGA-1 increases lifespan via maintaining mitochondrial fusion. Neuronal RAGA-1 abrogation of *raga-1* mutant longevity requires UNC-64/syntaxin, and promotes mitochondrial fission cell nonautonomously. Finally, deleting the mitochondrial fission factor DRP-1 renders the animal refractory to the pro-aging effects of neuronal RAGA-1. Our results highlight a new role for neuronal TORC1 in cell nonautonomous regulation of longevity, and suggest TORC1 in the central nervous system might be targeted to promote healthy aging.
DOI: https://doi.org/10.7554/eLife.49158.001

**\*For correspondence:**
wmair@hsph.harvard.edu

**Competing interests:** The authors declare that no competing interests exist.

## Introduction

Aging is the single biggest risk factor for the majority of non-communicable complex diseases, including some of those with the greatest negative impact on human health outcomes worldwide (*Escoubas et al., 2017*). Work over the last two decades has uncovered molecular mechanisms that can be manipulated in model organisms to modulate the aging process and reduce overall disease risk in old age (*Fontana et al., 2010*). Many of these interventions have been linked to nutrient and energy sensing pathways, whose modulation genetically or pharmacologically mimics the effects of dietary restriction on healthy aging (*Fontana and Partridge, 2015*). One classic example of a nutrient sensor linked to longevity is TORC1, which promotes anabolic processes such as protein translation to provide macromolecules for growth and proliferation while inhibiting catabolic activities such as autophagy (*Albert and Hall, 2015*). TORC1 is activated by growth factors and amino acids, the latter of which act through sensors such as the sestrins to facilitate heterodimer formation of the Rag proteins (consisting of RagA, B, C and D in mammals), an essential step for TORC1 activation (*Saxton and Sabatini, 2017*). Suppression of TORC1 both genetically and pharmacologically, via rapamycin feeding, promotes longevity in multiple species from yeast to mice (*Kennedy and Lamming, 2016*). In contrast to TORC1, the conserved kinase AMPK is activated under low energy conditions. AMPK activation promotes catabolic processes that generate ATP, including the TCA cycle, fatty acid oxidation and autophagy (*Burkewitz et al., 2014*), and extends lifespan in *C. elegans*

(*Apfeld et al., 2004*; *Greer et al., 2007*; *Mair et al., 2011*) and fruit flies (*Stenesen et al., 2013*). Given the antagonistic roles of TORC1 and AMPK in nutrient-sensing, metabolism and longevity, it is perhaps intuitive that they are mechanistically linked. Indeed, in mammals, AMPK is generally regarded as an upstream suppressor of TORC1 as it phosphorylates the TSC complex and raptor to inhibit TORC1 activity (*Gwinn et al., 2008*; *Inoki et al., 2003*). Conversely, TORC1 signaling has been suggested to inhibit AMPK via p70 ribosomal protein S6 Kinase (S6K) (*Dagon et al., 2012*). However, much of the work elucidating how AMPK and TORC1 interact has been performed in vitro in cell culture studies and it remains unclear in multicellular organisms whether TORC1 and AMPK causally coordinate to modulate the aging process.

Here, we elucidate the relationship between AMPK and TORC1 in the modulation of aging by discovering a critical role for neuronal AMPK in lifespan extension resulting from suppression of TORC1 components in *C. elegans*. We show that neuronal TORC1 pathway activity itself is critical for healthy aging. Restoring either *raga-1* or *rsks-1* (encoding the *C. elegans* orthologue of S6 Kinase) only in neurons fully suppresses lifespan extension in animals lacking the respective TORC1 component in all other cell types. However, the downstream effectors of AMPK longevity and TORC1 longevity appear separable; constitutive activation of CREB regulated transcriptional coactivator 'CRTC'−1, which suppresses AMPK longevity, does not suppress lifespan extension by TORC1 suppression. We show that neuronal RAGA-1 regulates metabolic genes in distal tissues and critically requires UNC-64/syntaxin exocytosis to modulate systemic longevity. Using RNA-Seq and in vivo reporters, we identify mitochondrial network state as the downstream mechanism for lifespan regulation by neuronal RAGA-1. Neuronal RAGA-1 activity regulates mitochondrial networks in distal tissues cell nonautonomously and this regulation is required for the effect of neuronal RAGA-1 on the aging process. Together, our data provide key insights into the tissue-specific roles of TORC1 suppression in healthy aging and highlight the nervous system as a putative target site for future pharmacological interventions aimed at the TORC1 pathway.

## Results

### AMPK is required for longevity mediated by TORC1 suppression

AMPK is canonically regarded as an upstream suppressor of TORC1. However, whether TORC1 suppression acts downstream of the pro-longevity effects of AMPK is unknown. Loss of the TORC1 target S6 Kinase increases lifespan in *C. elegans* and mouse (*Selman et al., 2009*) and results in transcriptional profiles in muscle that resemble those of active AMPK (*Selman et al., 2009*). Interestingly, AMPK activity is required for lifespan extension by loss of S6K in *C. elegans* (*Selman et al., 2009*), which does not support a model where AMPK acts linearly upstream of TORC1 to modulate aging.

S6K deletion has been shown to activate AMPK in both *C. elegans* and mice (*Selman et al., 2009*). To examine whether the activation of AMPK is specific to RSKS-1 (S6K) or pertains more broadly to the TORC1 pathway, we assayed the phosphorylation status of AMPK upon inhibition of multiple TORC1 components in *C. elegans*. We measured AMPK activity, using antibodies targeting phosphorylation of the α catalytic subunit at threonine 172, a site in the activation loop that is a hallmark of AMPK activation (*Hardie et al., 2012*). This phosphorylation site is conserved in *C. elegans* on the two AMPKα subunits, AAK-1 and AAK-2 (*Apfeld et al., 2004*) and its phosphorylation has been used to measure AMPK activity (*Selman et al., 2009*). To confirm conservation, we generated animals carrying a CRISPR induced threonine-to-alanine mutation at T243 on AAK-2, the residue homologous to mammalian T172. Lysates from both CRISPR generated AAK-2(T243A) mutation animals, and *aak-2(ok524)* mutants which harbor a 408 bp deletion spanning T243, completely lose recognition by antibody targeting phosphorylated AMPK T172 (*Figure 1—figure supplement 1a*), confirming conservation of this phosphorylation event (hereafter referred to as AMPK T172 phosphorylation) and specificity of the antibody to AAK-2. RNAi against *let-363* (homolog of TOR) significantly increases AMPK T172 phosphorylation (*Figure 1—figure supplement 1b,c*) in whole worm lysates. In addition, we found significantly increased levels of AMPK T172 phosphorylation in animals carrying a null allele in *raga-1* (*Figure 1a,b*), a gene encoding a homologous protein to RagA, which senses amino acids and activates TORC1 (*Saxton and Sabatini, 2017*). Taken together, these data suggest that in *C. elegans*, AMPK is activated when TORC1 signaling is decreased.

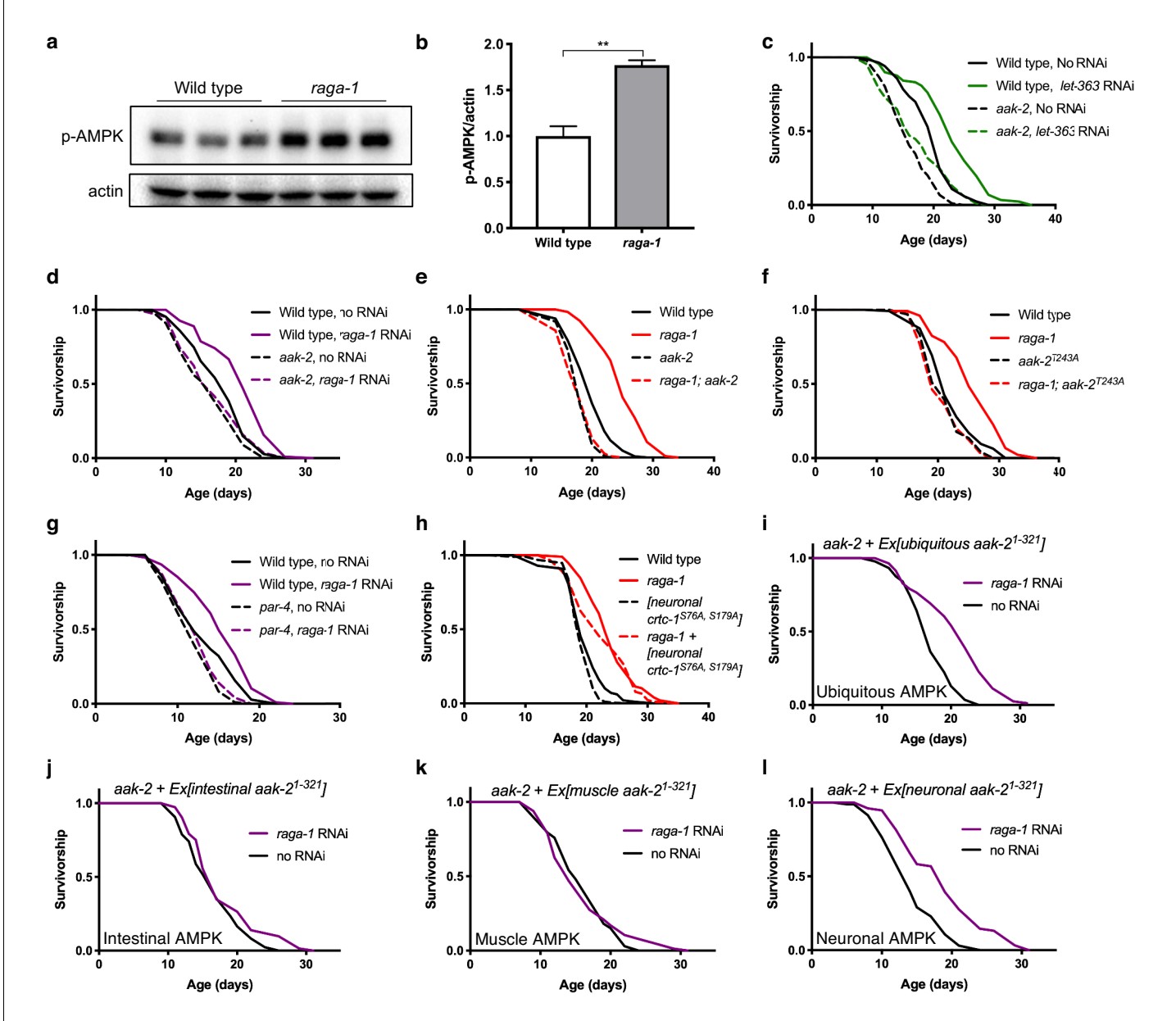

**Figure 1.** Neuronal AMPK is required for TORC1-mediated longevity. (a, b) AMPK T172 phosphorylation is increased in *raga-1(ok386)* null mutants compared to wild type animals. Actin levels are used as loading controls (representative immunoblot and corresponding quantification, n = 3 independent blots). Source data for *Figure 1b* are provided in *Figure 1—source data 1*. (c, d) Knockdown of (c) *let-363 (TOR)* (n = 3 independent biological replicates) or (d) *raga-1* from day 1 of adulthood extends lifespan in wild type animals (p<0.0001), but not in *aak-2(ok524)* null mutants (n = 8 independent biological replicates). *P* values: wild type on no RNAi vs *let-363* RNAi (p<0.0001), *aak-2(ok524)* on no RNAi vs *let-363* RNAi (p=0.0122), wild type vs *aak-2(ok524)* on *let-363* RNAi (p<0.0001); wild type on no RNAi vs *raga-1* RNAi (p<0.0001), *aak-2(ok524)* on no RNAi vs *raga-1* RNAi (p=0.1968). Sample size ranges between 62–115 deaths per treatment each replicate. *P* values are calculated with Log-rank (Mantel-Cox) test. Details on strains and lifespan replicates can be found in *Supplementary file 6*. (e) *raga-1(ok386)* mutants live longer than wild type (p<0.0001). *raga-1(ok386); aak-2(ok524)* double mutant animals do not live significantly longer than *aak-2(ok524)* single mutants (p=0.9735). n = 5 independent biological replicates, sample size ranges between 90–129 deaths per treatment each replicate. *P* values are calculated with Log-rank (Mantel-Cox) test. Details on strains and lifespan replicates can be found in *Supplementary file 6*. (f) *raga-1(ok386)* mutants live longer than wild type (p<0.0001). *raga-1(ok386)* does not extend lifespan in an *aak-2(wbm20)* [AAK-2 (T243A)] CRISPR-generated allele background (p=0.5354). n = 3 independent biological replicates, sample size ranges between 93–119 deaths per treatment each replicate. *P* values are calculated with Log-rank (Mantel-Cox) test. Details on strains and lifespan replicates can be found in *Supplementary file 6*. (g) At the restrictive temperature 22.5°C to induce *par-4* loss-of-function, *raga-1* RNAi from adulthood extends lifespan in wild type animals (p<0.0001) but does not extend lifespan of *par-4(it57)* mutant animals (p=0.206). Worms were grown at 15°C until L4 stage before shifting to restrictive temperature. n = 3 independent biological replicates, sample size 68–115 deaths per treatment each replicate. *P* values are

*Figure 1 continued on next page*

*Figure 1 continued*

calculated with Log-rank (Mantel-Cox) test. Details on strains and lifespan replicates can be found in *Supplementary file 6*. (h) *raga-1(ok386)* null allele extends lifespan in animals expressing the non-phosphorylatable CRTC-1$^{S76A, S179A}$ driven by a pan-neuronal *rab-3* promoter (p<0.0001). n = 3 independent biological replicates; sample size ranges between 90–121 deaths per treatment each replicate. *P* values are calculated with Log-rank (Mantel-Cox) test. Details on strains and lifespan replicates can be found in *Supplementary file 6*. (i) *raga-1* RNAi from adulthood extends median lifespan of animals overexpressing *aak-2*$^{aa1-321}$ driven by the ubiquitous *sur-5* promoter in *aak-2* mutant background (p<0.0001). n = 3 independent biological replicates; sample size ranges between 78–109 deaths per treatment. *P* values are calculated with Log-rank (Mantel-Cox) test. Details on strains and lifespan replicates can be found in *Supplementary file 6*. (j, k) *raga-1* RNAi from adulthood does not extend median lifespan of animals overexpressing *aak-2* $^{aa1-321}$ driven by (j) an intestine-specific *gly-19* promoter (p=0.0342) or (k) a muscle-specific *myo-3* promoter in *aak-2* mutant background (p=0.9827). n = 3 independent biological replicates; sample size ranges between 72–102 (intestine) and 44–107 (muscle) deaths per treatment each replicate. *P* values are calculated with Log-rank (Mantel-Cox) test. Details on strains and lifespan replicates can be found in *Supplementary file 6*. (l) *raga-1* RNAi from adulthood can extend median lifespan of animals overexpressing *aak-2* $^{aa1-321}$ driven by the pan-neuronal *rab-3* promoter in an *aak-2* mutant background (p<0.0001). n = 4 independent biological replicates, sample size ranges between 66–117 deaths per treatment each replicate. *P* values are calculated with Log-rank (Mantel-Cox) test. Details on strains and lifespan replicates can be found in *Supplementary file 6*.

DOI: https://doi.org/10.7554/eLife.49158.002

The following source data and figure supplements are available for figure 1:

**Source data 1.** *Figure 1b* AMPK activity is increased in raga-1 mutants.
DOI: https://doi.org/10.7554/eLife.49158.004
**Figure supplement 1.** Conservation of critical residues in AAK-2 and their requirement for TORC1-mediated longevity.
DOI: https://doi.org/10.7554/eLife.49158.003
**Figure supplement 1—source data 1.** *Figure 1—figure supplement 1c* AMPK activity is increased by knockdown of TOR.
DOI: https://doi.org/10.7554/eLife.49158.005
**Figure supplement 1—source data 2.** *Figure 1—figure supplement 1f* The conserved S6K/Akt phosphorylation site serine 551 on AAK-2 modulates AMPK activity.
DOI: https://doi.org/10.7554/eLife.49158.006

We next tested whether AMPK activity is required for longevity resulting from TORC1 suppression. Adult-onset RNAi of *let-363* (TOR) or *raga-1* extended median lifespan by 20% in wild type animals, but this effect is suppressed in *aak-2(ok524)* mutants (*Figure 1c,d*). In addition, while null mutation of *raga-1* extends lifespan by 30% in an otherwise wild type background, it has no effect in *aak-2(ok524)* mutants (*Figure 1e*). Supporting the hypothesis that phosphorylation of AMPK at T172 is critical for extension of lifespan by TORC1 suppression, the extended lifespan of *raga-1* mutants was also suppressed by AAK-2 T243A (*Figure 1f*). These results are unlikely due to the general sickness of the *aak-2* mutants, since their lifespan can be extended by other interventions, such as dietary restriction in liquid culture or intermittent fasting (*Greer and Brunet, 2009*; *Honjoh et al., 2009*; *Mair et al., 2009*). LKB1 is the primary kinase that phosphorylates AMPK under energy stress (*Tsou et al., 2011*). We tested the requirement for PAR-4, the LKB1 homolog in *C. elegans*, for *raga-1* mediated longevity using animals bearing a temperature sensitive allele *par-4(it57)* (*Morton et al., 1992*). *par-4(it57)* animals are refractory to the lifespan extension of *raga-1* RNAi when housed at restrictive temperatures, suggesting AMPK T172 phosphorylation by LKB1 is required for TORC1-mediated longevity (*Figure 1g*).

S6K has been suggested to phosphorylate AMPK alpha 2 at serine 485, resulting in inhibition of AMPK activity (*Dagon et al., 2012*). We examined whether phosphorylation status at the *C. elegans* equivalent of AMPK alpha 2 serine 485 mediates lifespan extension by TORC1 pathway suppression. Although we found that this serine residue is conserved in *C. elegans* (serine 551 residue in AAK-2) (*Figure 1—figure supplement 1d*) and inhibits AMPK (*Figure 1—figure supplement 1e,f*), mutating S551 to alanine does not alter the longevity response to reduced TORC1 signaling (*Figure 1—figure supplement 1g*), suggesting that this phosphorylation event is not required for TORC1-mediated longevity. Taken together, these data suggest that in *C. elegans*, AMPK is activated and critically required for lifespan extension when TORC1 is inhibited.

## TORC1 and AMPK act via separable downstream mechanisms

If AMPK acts as the key downstream effector of TORC1 mediated longevity, we reasoned that interventions that inhibit lifespan extension via AMPK activation might also block longevity resulting from suppression of TORC1. Previously, we have shown that phosphorylation of CREB-regulated

transcriptional coactivator (CRTC)−1 in neurons is required for lifespan extension by constitutively activated AMPK (*Burkewitz et al., 2015*). To determine if lifespan extension via *raga-1* deletion acts via a similar mechanism, we crossed a transgene that expresses a non-phosphorylatable variant of CRTC-1[(S76A, S179A)] only in neurons into *raga-1(ok386)* mutants. Although neuronal CRTC-1[(S76A, S179A)] fully suppresses AMPK mediated longevity (*Burkewitz et al., 2015*), it does not similarly suppress *raga-1* mutant longevity (*Figure 1h*). Together these data suggest that, while AMPK is required for TORC1 mediated longevity, the mechanisms by which TORC1 suppression and AMPK activation promote healthy aging are likely separable.

## Neuronal AMPK is required for TORC1-mediated longevity

To further explore why AMPK activity is required for lifespan extension by TORC1 suppression, we asked where AMPK acts to mediate this effect. We rescued *aak-2(ok524)* mutants in specific tissues with a truncated and constitutively active form of AAK-2[aa1-321] (*Mair et al., 2011*). Ubiquitous expression of AAK-2[aa1-321] fully restored the responsiveness of *aak-2(ok524)* mutant animals to the lifespan promoting effects of *raga-1* RNAi (*Figure 1i*), whereas expression of AAK-2[aa1-321] specifically in either the intestine or muscle did not (*Figure 1j,k*). Remarkably, however, expressing AAK-2[aa1-321] solely in neurons fully restored lifespan extension by *raga-1* RNAi (*Figure 1l*). We saw similar rescue using neuronal expression of full-length wild type *aak-2* (*Figure 1—figure supplement 1h*). These data demonstrate that activity of AMPK specifically in neurons is a critical mediator of TORC1 longevity, and also suggest that neuronal TORC1 activity itself may impact healthy aging.

## TORC1 acts in neurons to regulate aging

Homozygous null mutations for many TORC1 components and regulators lead to developmental arrest (*Albert and Riddle, 1988*; *The C. elegans Deletion Mutant Consortium, 2012*; *Long et al., 2002*) and therefore do not facilitate longevity assays. However, null mutants for *raga-1* or *rsks-1* are viable and long-lived (*Schreiber et al., 2010*; *Selman et al., 2009*). To directly examine the role of neuronal TORC1 activity on organismal aging, we rescued TORC1 pathway activity in *raga-1* and *rsks-1* mutants specifically in neurons, via ectopic expression regulated by the pan-neuronal *rab-3* promoter. These animals therefore have reduced TORC1 signaling in all tissues except neurons. Strikingly, we found that expression of *raga-1* in neurons fully suppressed the long lifespan of *raga-1* mutant animals without affecting wild type lifespan (*Figure 2a*). Neuronal *raga-1* fully suppressed the long lifespan both with and without the use of FUDR, a chemical inhibitor of DNA synthesis commonly used in *C. elegans* lifespan experiments to stop the production of progeny that can also lead to complex interactions with various genes and treatments in the regulation of lifespan (*Anderson et al., 2016*) (*Figure 2—figure supplement 1*). Similarly, neuronal *rsks-1* expression rescued the lifespan extension of *rsks-1* deletion mutants (*Figure 2b*), suggesting that neuronal regulation of longevity is not specific to RAGA-1 and extends to other TORC1 pathway components.

Since many neuronal-specific promoters show leakiness in the intestine, especially in adult *C. elegans*, we used an RNAi protocol to eliminate transgene expression in the intestine while leaving neuronal expression intact (*Figure 2—figure supplement 2a*). Critically, in animals in which intestinal leakiness of the promoter was removed via *raga-1* RNAi, *rab-3p::raga-1* still fully suppressed *raga-1* mutant lifespan (*Figure 2—figure supplement 2b*). Additionally, we generated single copy knockins of *raga-1* cDNA using the SKI LODGE system (*Silva-García et al., 2019*). We used CRISPR to insert *raga-1* cDNA sequences into an intergenic cassette which uses both the *rab-3* promoter and *rab-3* 3' UTR sequences to drive gene expression more specifically in the nervous system (*Silva-García et al., 2019*). We also knocked *raga-1* cDNA into a cassette that uses the *eft-3* promoter to express *raga-1* across all somatic tissues. Remarkably, *raga-1* expressed in the nervous system by single copy transgene effectively rescued *raga-1* lifespan, despite its more restricted and lower level of expression (*Figure 2c,d*). In contrast, expressing *raga-1* directly in the intestine by multicopy array using the intestine-specific *ges-1* promoter did not suppress *raga-1* mutant longevity (*Figure 2e*). Expression of *raga-1* in the nervous system by single copy transgene did not rescue other phenotypes resulting from loss of RAGA-1 function, including reduced body size (*Figure 2f*) and developmental delay (*Figure 2—figure supplement 3*). Together, these results show that TORC1 activity specifically in neurons modulates systemic aging, and this activity can be uncoupled from its regulation of growth and development.

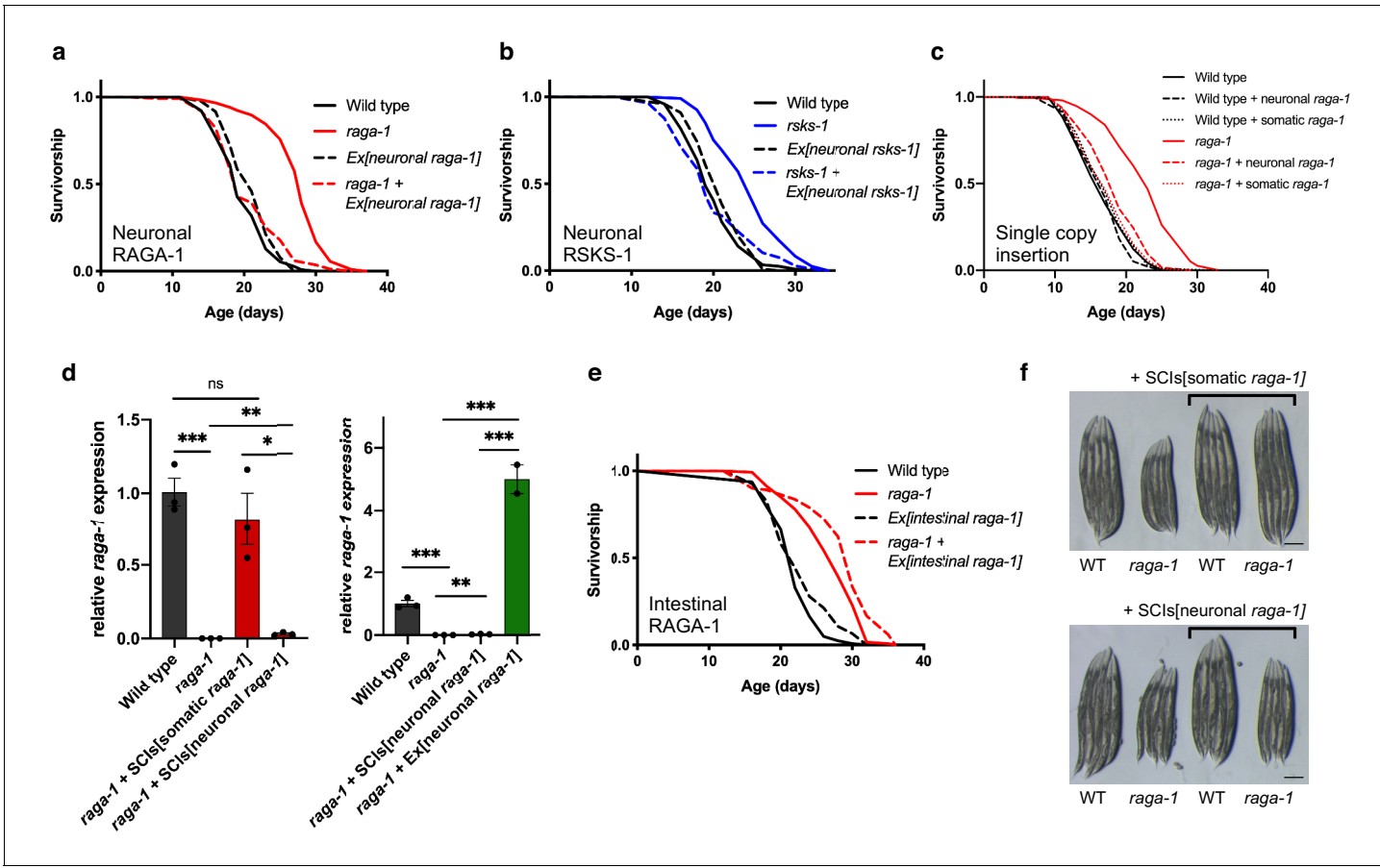

**Figure 2.** TORC1 signaling is required in neurons to regulate lifespan. (a) The *raga-1(ok386)* deletion increases lifespan (p<0.0001). However, when *raga-1* is expressed in the nervous system via extrachromosomal array using the *rab-3* promoter, *raga-1(ok386)* does not extend lifespan (p=0.7932). n = 6 independent biological replicates; sample size ranges between 77–192 deaths per treatment each replicate. *P* values are calculated with Log-rank (Mantel-Cox) test. Details on strains and lifespan replicates can be found in *Supplementary file 6*. (b) The *rsks-1(ok1255)* deletion increases lifespan (p<0.0001). However, when *rsks-1* is expressed in the nervous system by extrachromosomal array using the *rab-3* promoter, *rsks-1(ok1255)* does not extend lifespan (p=0.6539). n = 4 independent biological replicates; sample size ranges between 81–121 deaths per treatment each replicate. *P* values are calculated with Log-rank (Mantel-Cox) test. Details on strains and lifespan replicates can be found in *Supplementary file 6*. (c) The lifespan extension resulting from *raga-1(ok386)* deletion is significantly suppressed by reintroduction of *raga-1* in the nervous system as a single copy transgene (p<0.0001). n = 3 independent biological replicates. *P* value calculated with Log-rank (Mantel-Cox) test. As described in Materials and methods, 'neuronal *raga-1*' refers to single copy transgene driven by the *rab-3* promoter. 'somatic *raga-1*' refers to single copy transgene driven by the *eft-3* promoter. Details on strains used and lifespan replicates can be found in *Supplementary file 6*. (d) Relative *raga-1* expression generated from integrated single copy transgenes (left) compared to extrachromosomal array (right) as determined by qPCR. Points plotted are from independent biological samples. Error bars denote mean + /- SEM. *P* values determined by 2-tailed t test. *raga-1* expression driven from single copy insertion transgenes (SCIs) and controls shown on the left were replotted next to expression generated by extrachromosomal array on the right to highlight relative differences in expression. All samples were run in parallel. (e) *raga-1(ok386)* allele extends lifespan both in the wild type background (p<0.0001) and with *raga-1* expressed in the intestine by extrachromosomal array using the intestine-specific *ges-1* promoter (p<0.0001). n = 3 independent biological replicates; sample size ranges between 83–126 deaths per treatment each replicate. *P* values are calculated with Log-rank (Mantel-Cox) test. Details on strains and lifespan replicates can be found in *Supplementary file 6*. (f) Somatic (top), but not neuronal (bottom), expression of *raga-1* driven by single copy transgene rescues the small body size exhibited by *raga-1(ok386)* mutants. Scale bar = 200 μm.

DOI: https://doi.org/10.7554/eLife.49158.007

The following source data and figure supplements are available for figure 2:

**Source data 1.** *Figure 2e* qPCR of raga-1 expression in SCIs and extrachromosomal lines.
DOI: https://doi.org/10.7554/eLife.49158.011

**Figure supplement 1.** Neuronal *raga-1* expressed by extrachromosomal transgene suppresses *raga-1* mutant longevity when animals are not treated with FUDR to prevent progeny development.
DOI: https://doi.org/10.7554/eLife.49158.008

**Figure supplement 2.** Elimination of *raga-1* in the intestine by RNAi does not impair rescue by the extrachromosomal neuronal *raga-1* array.
DOI: https://doi.org/10.7554/eLife.49158.009

*Figure 2 continued*

**Figure supplement 3.** Neuronal *raga-1* does not rescue development delay of *raga-1(ok386)* mutants.

DOI: https://doi.org/10.7554/eLife.49158.010

**Figure supplement 3—source data 1.** *Figure 2—figure supplement 3* Developmental stages of raga-1 rescue lines.

DOI: https://doi.org/10.7554/eLife.49158.012

## Neuronal RAGA-1 modulates aging via neuropeptide signaling

To identify mechanisms specifically coupled to neuronal RAGA-1 regulation of lifespan, we examined the transcriptomes of wild type ('WT'), *raga-1* mutant ('mutant') and *raga-1* mutant with extrachromosomal neuronal *raga-1* expression ('rescue') *C. elegans* by RNA-Seq (*Supplementary files 1–3*). We performed a cluster analysis that takes into account trends in gene expression across all three conditions to identify genes that change in the *raga-1* mutant and are reversed by neuronal rescue. As expected given the specific nature of neuronal *raga-1* rescue of lifespan but not other *raga-1* phenotypes, the majority of differentially expressed genes changed similarly in the *raga-1* mutant and in the rescue line (*Figure 3—figure supplement 1a*). However, a small number of genes that showed increased or decreased expression in the *raga-1* mutant were rescued by the neuronal *raga-1* array (*Figure 3a*, *Figure 3—figure supplement 1a*). Interestingly, functional analysis of the genes within these clusters reveals an enrichment for Gene Ontology (GO) terms related to organelle organization, organelle fission, and unfolded protein/ER stress pathways that are upregulated in the *raga-1* mutant but not in rescue, whereas GO terms related to neuronal function, including synaptic structure and signaling as well as regulation of dauer entry are enriched in the cluster of genes that show reduced expression in *raga-1* but not in rescue (*Figure 3b*). These GO terms are also revealed in pairwise comparisons designed to identify biological processes that differ between wild type and *raga-1*, but not between wild type and neuronal rescued animals (*Figure 3—figure supplement 1b, c*).

Among the genes represented in Cluster 4 (decreased in *raga-1*, increased in rescue) are multiple insulin-like peptide genes (*daf-28*, *ins-6*, *ins-26*, *ins-30*) including two, *daf-28* and *ins-6*, that when overexpressed are sufficient to suppress the lifespan extension resulting from loss of chemosensory function (*Artan et al., 2016*) (*Supplementary file 2*). We verified increased expression of *daf-28* and *ins-6*, as well as other identified changes in gene expression, in L4 animals rescued either by extrachromosomal array or by single copy transgene knock-in by qPCR (*Figure 3c,d*, *Figure 3—figure supplement 2*). In adults, the levels of *ins-6* expression induced by neuronal or somatic expression of *raga-1* are indistinguishable, suggesting its regulation is entirely neuronal (*Figure 3—figure supplement 3*). These data suggest that neuronal TORC1 might mediate systemic longevity cell nonautonomously via secretion of insulin-like peptides or other *raga-1* dependent neuronal signals.

Neuropeptides are released from dense core vesicles (DCVs) (*Li, 2008*). If neuronal RAGA-1 suppresses *raga-1* longevity via expression of insulin-like peptides or other neuropeptide signals, we reasoned the lifespan of *raga-1* neuronal rescue worms might be de-repressed by blocking neuropeptide release. We utilized mutants for *unc-64*, a homolog of mammalian syntaxin, an essential plasma membrane receptor for DCV exocytosis to test whether impairing neuronal function in this way would block the ability of the neuronal extrachromosomal *raga-1* array to rescue. Hypomorphic *unc-64(e246)* mutant animals are defective for dense core vesicle docking (*Zhou et al., 2007*) and remarkably completely remove suppression of *raga-1* lifespan by neuronal RAGA-1. *raga-1; unc-64* double mutant animals with the neuronal rescue array live more than 40% longer than *raga-1* neuronal rescue animals (*Figure 3e*). The longevity effects by *unc-64* mutation on *raga-1* neuronal rescue animals, combined with the RNA seq results, strongly suggest that neuronal TORC1 actively causes the release of neuropeptide signals to limit longevity cell nonautonomously.

## Neuronal RAGA-1 drives peripheral mitochondrial fragmentation in aging animals

Since genes linked to organelle organization were upregulated in the *raga-1* mutant but not in rescue, we explored whether this might be causal to longevity of the *raga-1* mutants.

Mitochondria can dynamically move between fused and fragmented networks in response to changes in the cellular environment, mediated in *C. elegans* by the GTPases: FZO-1 (fusion) and

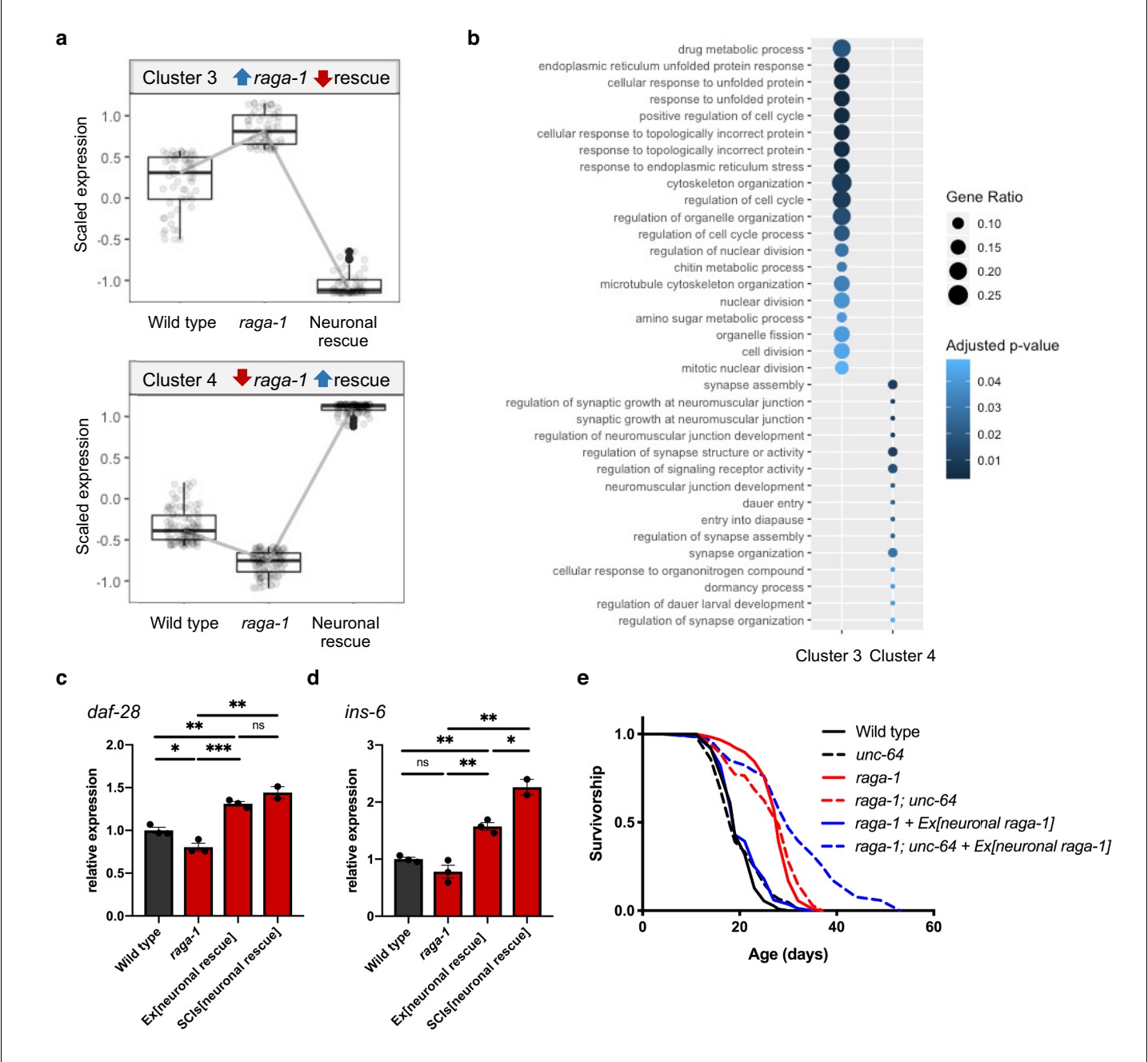

**Figure 3.** Neuronal TORC1 modulates aging via changes to organelle organization and neuropeptide signaling. (a) Cluster analysis identified 59 genes that show increased expression in *raga-1(ok386)* that is reversed by neuronal rescue array (Cluster 3, top) and 107 genes that show decreased expression in *raga-1* that is reversed by neuronal rescue (Cluster 4, bottom). (b) Over-representation analysis of the Gene Ontology biological process terms for genes comprising each of the clusters shown in panel (a). (c, d) qPCR to validate increased expression of *daf-28* (c) and *ins-6* (d) in independent samples. Samples were generated from L4 larvae to match the timepoint of RNA-Seq sample collection. Independent biological replicates were processed and amplified in parallel. 'Ex[neuronal rescue]' refers to *raga-1(ok386)* animals rescued by extrachromosomal array, 'SCIs [neuronal rescue]' refers to *raga-1(ok386)* animals rescued by single copy insertion. *P* values determined by 2-tailed t test. Source data are provided in **Figure 3—source data 1**. (e) The *unc-64(e246)* hypomorphic allele significantly extends lifespan in *raga-1* neuronal rescue background (p<0.0001), but not in wild type (p=0.3449) or *raga-1* mutant (p=0.2708). n = 3 independent biological replicates, sample size ranges between 64–121 deaths per treatment each replicate. *P* values are calculated with Log-rank (Mantel-Cox) test. Details on strains and lifespan replicates can be found in **Supplementary file 6**.

DOI: https://doi.org/10.7554/eLife.49158.013

The following source data and figure supplements are available for figure 3:

*Figure 3 continued on next page*

*Figure 3 continued*

**Source data 1.** *Figure 3c and d* qPCR of daf-28 and ins-6.
DOI: https://doi.org/10.7554/eLife.49158.017
**Figure supplement 1.** Gene clusters and differentially represented GO terms identified by analysis of RNA-seq.
DOI: https://doi.org/10.7554/eLife.49158.014
**Figure supplement 2.** Validation of changes identified by RNA-seq in independent biological samples.
DOI: https://doi.org/10.7554/eLife.49158.015
**Figure supplement 2—source data 1.** *Figure 3—figure supplement 2* qPCR validation of RNA seq results.
DOI: https://doi.org/10.7554/eLife.49158.018
**Figure supplement 3.** Neuronal *raga-1* regulates *ins-6* expression in adults.
DOI: https://doi.org/10.7554/eLife.49158.016
**Figure supplement 3—source data 1.** *Figure 3—figure supplement 3* qPCR of ins-6 in day 1 adults.
DOI: https://doi.org/10.7554/eLife.49158.019

DRP-1 (fission) (*Wai and Langer, 2016*). We examined mitochondrial networks in young and old *C. elegans* in multiple tissues using reporters expressing GFP fused with a fragment from the mitochondrial outer membrane protein TOMM-20, which includes a transmembrane domain that anchors the fusion protein to mitochondria (*Weir et al., 2017*). For neurons and muscle, we developed an ImageJ/FIJI macro, 'MitoMAPR', to characterize and quantify changes in mitochondrial architecture and morphology in *C. elegans* cells. MitoMAPR uses several pre-existing image enhancement filters to augment the signal clarity of images, converting the mitochondrial signals into skeleton-like binary backbones (*Figure 4—figure supplement 1*), and quantifying various attributes of the mitochondrial network as described in (*Figure 4—figure supplement 2*). Coupled with a high-throughput batch processing mode, this macro allows us to characterize changes in mitochondrial object length, distribution, network coverage and complexity in a large number of images acquired from the samples in question (*Figure 4*, *Figure 4—figure supplement 3*, *Figure 4—figure supplement 5*). Age induces fragmentation of mitochondria in neurons, intestine, and muscle, in wild type animals but degeneration is not seen in long lived *raga-1* mutants (*Figure 4*, *Figure 4—figure supplement 3*, *Figure 4—figure supplement 4*, *Figure 4—figure supplement 5*).

To test whether mitochondria in peripheral tissues might be affected by signals generated by neuronal TORC1 activity cell nonautonomously, we examined mitochondrial morphology in body wall muscle in wild type, *raga-1* mutant and *raga-1* neuronal rescue animals (*Figure 4a,b*). Mitochondrial networks in wild type muscle cells showed decreased total coverage (measured by percentage of cell area covered by mitochondria) (*Figure 4c*) and size (measured by length and area of each mitochondria particle) with age (*Figure 4d,e*). Further suggesting an increase in fragmentation of mitochondria in WT muscle cells with age as seen previously (*Weir et al., 2017*), we saw an increased object number normalized to area (*Figure 4f*). Strikingly, loss of *raga-1* attenuates both the age-related decline in mitochondrial coverage and the increase in fragmentation (*Figure 4c,d,e, f*). Interestingly, the mitochondrial architecture in *raga-1* mutants contains fewer networks and junction points compared to WT at all ages tested, suggesting that mitochondria are longer but less interconnected (*Figure 4—figure supplement 5*). Next, we examined the impact of neuronal *raga-1* on mitochondria in muscle cells. *raga-1* neuronal rescue suppressed the effect of *raga-1* deletion on muscle mitochondria in both young and old animals, causing reduced coverage, decreased size and a higher degree of fragmentation (*Figure 4c,d,e,f*). Further, network counts in *raga-1* neuronal rescue animals are not significantly different to wild type in young or old animals (*Figure 4—figure supplement 5*). Together, these data suggest that loss of *raga-1* specifically in neurons can maintain youthful mitochondrial network states with age in non-neuronal tissues.

Next, we examined the effect of loss of UNC-64/syntaxin function, which de-represses the effects of neuronal TORC1 on longevity, on muscle mitochondria (*Figure 4—figure supplement 6a,b*). In young *unc-64* mutants, mitochondria coverage and area were decreased in muscle cells and the networks show a higher degree of fragmentation compared to wild type. However, in *unc-64* mutants, unlike for WT, neither mitochondrial coverage nor area show significant alterations with age (*Figure 4—figure supplement 6c,d,e,f*). Taken together, our data suggest that neuronal signaling driven by TORC1 can cell nonautonomously drive mitochondrial fragmentation in peripheral tissues.

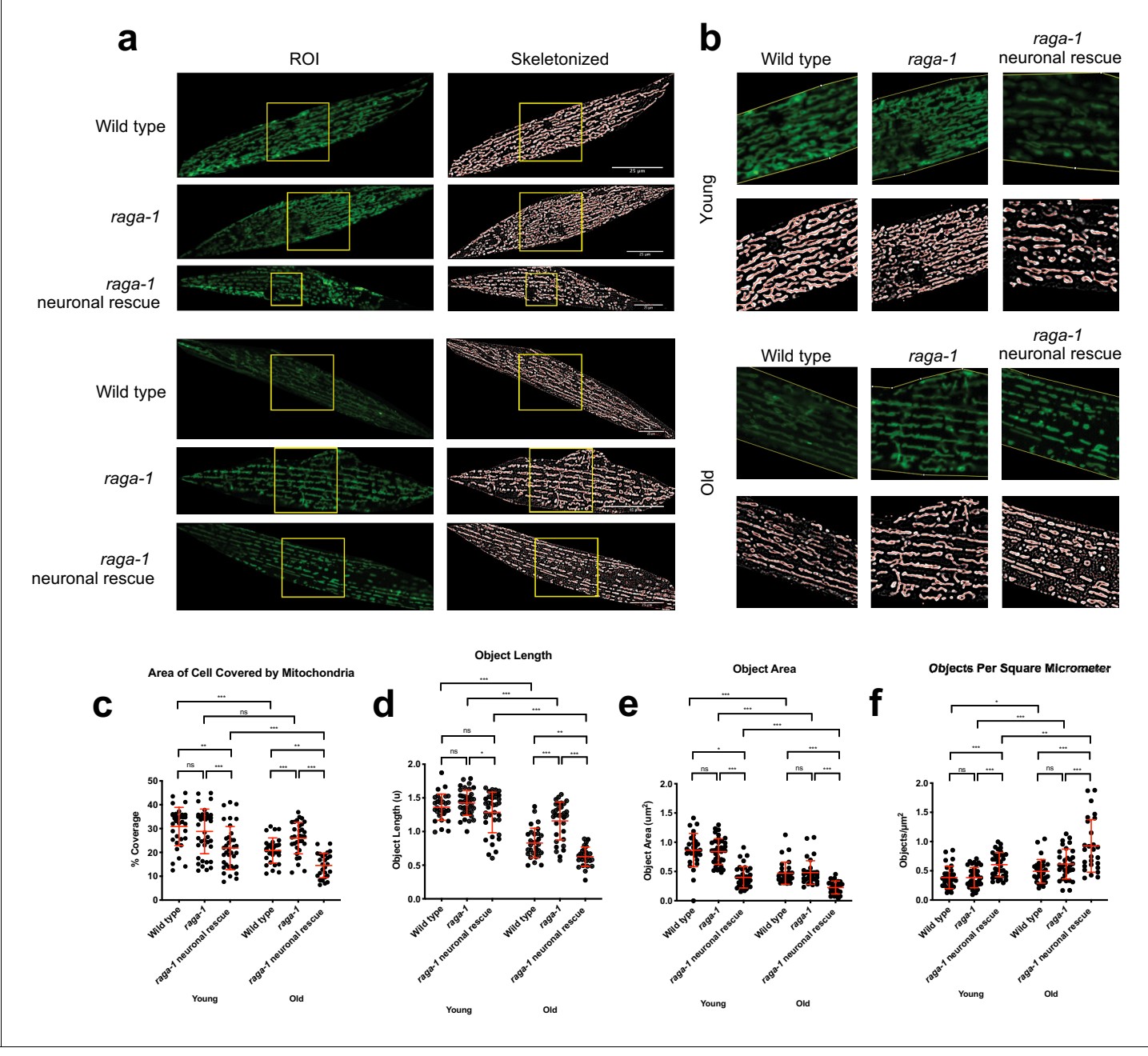

**Figure 4.** Neuronal RAGA-1 drives mitochondrial fragmentation in muscle cells. (a) Representative pictures showing that loss of *raga-1* preserves muscle mitochondrial content during aging, while neuronal RAGA-1 reverses these effects as seen in the corresponding skeletonized images (Left) post MitoMAPR processing. The mitochondrial skeleton (red) is overlaid on binary images. TOMM-20$^{aa1-49}$::GFP reporter labels mitochondria in young (day 2) and old (day 11) wild type, *raga-1(ok386)* mutant and *raga-1* neuronal rescue animals (n = 2 independent trials, 16–29 cells were imaged per genotype each time point per replicate). Scale bar represents 25 μm. (b) zoomed insets from (a) (yellow boxes). (c–h) Quantification showing that neuronal *raga-1* rescue animals also have decreased mitochondrial coverage (c), greater degree of fragmentation as seen by decreased mitochondrial length (d) and area (e) and increased object number normalized to area (f) compared to *raga-1* mutants. Data are represented as mean ±S.D. P value: NS no significance, *<0.05, **<0.01, ***<0.001, between comparisons as indicated by bars. Statistical significance was determined by One-way ANOVA and Welch's t test. 16–29 cells were quantified per genotype for each time point per replicate (n = 2 independent trials, 25–30 cells were imaged per genotype each time point per replicate). Source data are provided in *Figure 4—source data 1*.

DOI: https://doi.org/10.7554/eLife.49158.020

The following source data and figure supplements are available for figure 4:

**Source data 1.** *Figure 4c-h* and *Figure 4—figure supplement 5* Effects of neuronal *raga-1* rescue on parameters of muscle mitochondrial morphology.

*Figure 4 continued on next page*

*Figure 4 continued*

DOI: https://doi.org/10.7554/eLife.49158.027

**Figure supplement 1.** Workflow for MitoMAPR analysis.

DOI: https://doi.org/10.7554/eLife.49158.021

**Figure supplement 2.** Examples of networks analyzed by MitoMAPR.

DOI: https://doi.org/10.7554/eLife.49158.022

**Figure supplement 3.** *raga-1* deletion affects mitochondrial network states in neurons.

DOI: https://doi.org/10.7554/eLife.49158.023

**Figure supplement 3—source data 1.** *Figure 4—figure supplement 3c-f* Effects of neuronal *raga-1* rescue on parameters of neuronal mitochondrial morphology.

DOI: https://doi.org/10.7554/eLife.49158.028

**Figure supplement 4.** *raga-1* deletion prevents mitochondria fragmentation in intestine.

DOI: https://doi.org/10.7554/eLife.49158.024

**Figure supplement 4—source data 1.** *Figure 4—figure supplement 4c* Network states of intestinal mitochondria in *raga-1* mutants.

DOI: https://doi.org/10.7554/eLife.49158.029

**Figure supplement 5.** Neuronal *raga-1* expression alters muscle mitochondrial architecture.

DOI: https://doi.org/10.7554/eLife.49158.025

**Figure supplement 6.** Effects of the *unc-64* hypomorphic allele on muscle mitochondria morphology.

DOI: https://doi.org/10.7554/eLife.49158.026

**Figure supplement 6—source data 1.** *Figure 4—figure supplement 6c-f* Mitochondria network characteristics of muscle mitochondria in *unc-64* mutants.

DOI: https://doi.org/10.7554/eLife.49158.030

## *raga-1* deletion specifically requires a fused mitochondrial network to promote longevity

We sought to determine whether the changes we observed in mitochondrial network state were causally associated with *raga-1* longevity. First, we asked whether *raga-1* mutant animals require a fused mitochondrial network to extend lifespan. We crossed the *raga-1(ok386)* mutants with animals carrying a null allele of *fzo-1*, which encodes a protein orthologous to mammalian mitofusins. *fzo-1 (tm1133)* mutant animals are therefore defective in mitochondrial fusion and consequently have fragmented mitochondria (*Breckenridge et al., 2008*; *Ichishita et al., 2008*). *fzo-1(tm1133)* significantly suppresses the lifespan extension seen in *raga-1* mutants, which suggests that mitochondrial fusion is required for *raga-1* mediated longevity (*Figure 5a*). Mutations in *fzo-1* suppress the long lifespan both with and without the use of FUDR (*Figure 5—figure supplement 1*). Deleting the *C. elegans* dynamin-related protein 1 (DRP-1), which is required for mitochondrial fission, does not block *raga-1 (ok386)* longevity (*Figure 5b*), indicating that *raga-1* longevity specifically requires mitochondria fusion. In addition to the previous data on the effects of neuronal CRTC-1, these data emphasize how AMPK and TORC1 modulate aging in *C. elegans* by distinct mechanisms: AMPK longevity requires both fusion and fission (*Weir et al., 2017*), while lifespan extension by in *raga-1* mutants specifically requires mitochondrial fusion.

We examined which tissues require fused mitochondrial networks to facilitate lifespan extension via *raga-1* deletion. We restored mitochondrial fusion specifically in neurons in *fzo-1(tm1133)* mutants via rescue of *fzo-1* cDNA with a pan-neuronal *rab-3* promoter. This neuronal rescue failed to restore lifespan extension by loss of *raga-1* (*Figure 5c*). Similarly, expression of *fzo-1* in muscles failed to restore lifespan extension (*Figure 5d*), suggesting that mitochondrial fusion in neither neurons nor muscle underpin lifespan extension by TORC1 suppression. In contrast however, restoring mitochondrial fusion in intestine by expressing *fzo-1* in the *fzo-1(tm1133)* mutants enabled *raga-1* deletion to significantly extend lifespan (*Figure 5e*). These data support the hypothesis that neuronal RAGA-1 activity modulates lifespan via cell nonautonomous regulation of mitochondrial dynamics in distal tissues.

## Neuronal RAGA-1 suppresses lifespan cell nonautonomously via mitochondrial fission

Finally, having shown that mitochondrial fusion is required for lifespan extension in *raga-1(ok386)* mutants, we tested whether neuronal rescue of *raga-1* suppresses lifespan via promoting peripheral

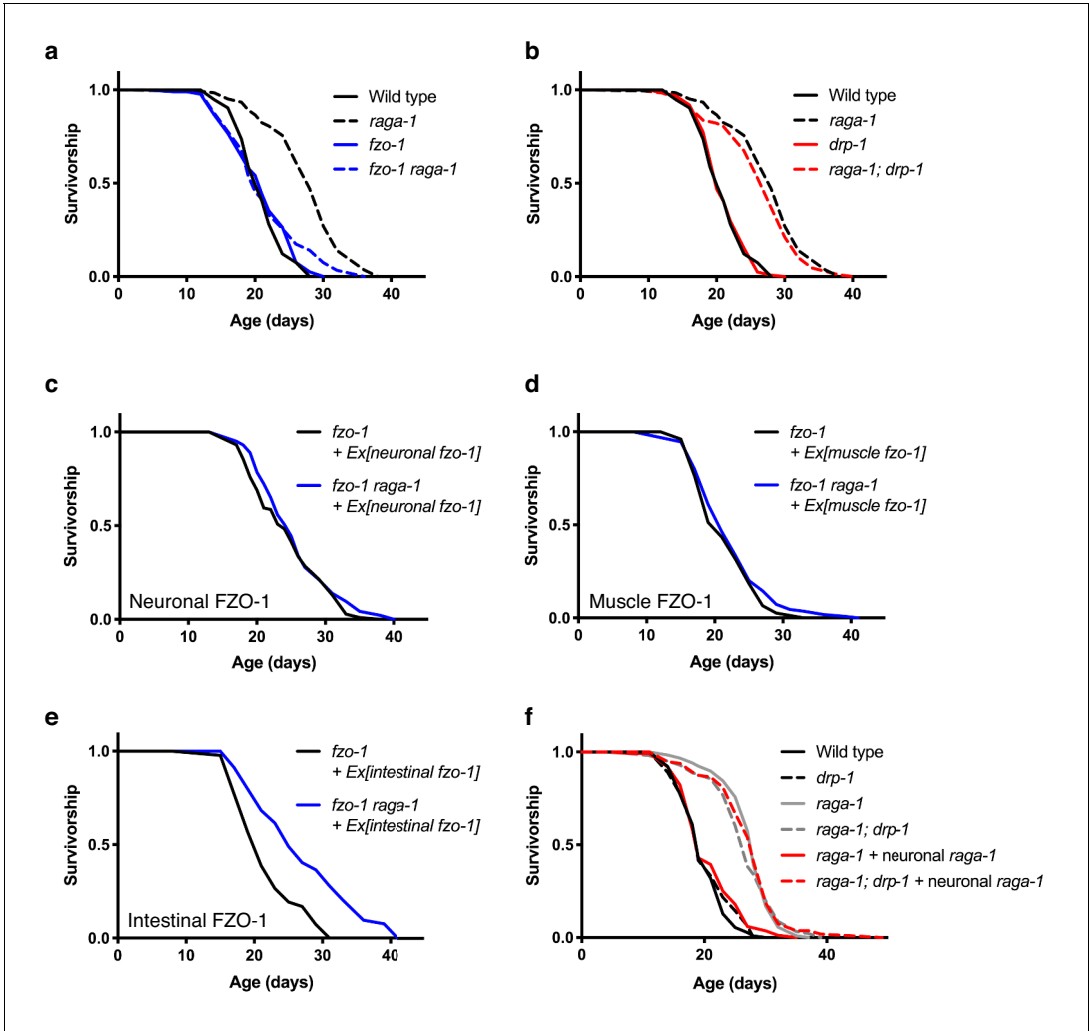

**Figure 5.** *raga-1* deletion requires a fused mitochondrial network to promote longevity. (a) *fzo-1(tm1133) raga-1(ok386)* double mutants, which are deficient in mitochondrial fusion, have significantly shortened lifespan compared to *raga-1(ok386)* single mutants (p<0.0001). However, *raga-1(ok386); drp-1(tm1108)* double mutants, which are deficient in mitochondrial fission, have similar lifespan compared to *raga-1(ok386)* single mutants (p=0.1420) (b). n = 4 independent biological replicates; sample size ranges between 71–128 deaths per treatment each replicate. Lifespan curves in a and b were generated in the same experiment but plotted separately for clarity, therefore wild type and *raga-1* control curves are shared by both. *P* values were calculated with Log-rank (Mantel-Cox) test. Details on strains and lifespan replicates can be found in *Supplementary file 6*. (c) When *fzo-1* is rescued in the *fzo-1(tm1133)* loss-of-function mutants using the neuronal-specific *rab-3* promoter, the lifespan extension by *raga-1* deletion is not fully restored (p=0.0428). n = 3 independent biological replicates; sample size ranges between 71–101 deaths per treatment each replicate. *P* values were calculated with Log-rank (Mantel-Cox) test. Details on strains and lifespan replicates can be found in *Supplementary file 6*. (d) When *fzo-1* is rescued in the *fzo-1 (tm1133)* loss-of-function mutants using the muscle-specific *myo-3* promoter, the lifespan extension by *raga-1* deletion is not restored (p=0.2011). n = 3 independent biological replicates; sample size ranges between 77–95 deaths per treatment each replicate. *P* values were calculated with Log-rank (Mantel-Cox) test. Details on strains and lifespan replicates can be found in *Supplementary file 6*. (e) When *fzo-1* is rescued in the *fzo-1(tm1133)* loss-of-function mutants using the intestinal-specific *ges-1* promoter, the lifespan extension by *raga-1* deletion is fully restored (p<0.0001). n = 3 independent biological replicates; sample size ranges between 43–104 deaths per treatment each replicate. *P* values were calculated with Log-rank (Mantel-Cox) test. Details on strains and lifespan replicates can be found in *Supplementary file 6*. (f) The *drp-1(tm1108)* deletion allele significantly extends lifespan in *raga-1* neuronal rescue background (p<0.0001), but not in wild type (p=0.3440) or *raga-1* mutant animals (p=0.4934). n = 4 independent biological replicates; sample size ranges between 78–120 deaths per treatment each replicate. *P* values are calculated with Log-rank (Mantel-Cox) test. Details on strains and lifespan replicates can be found in *Supplementary file 6*.

DOI: https://doi.org/10.7554/eLife.49158.031

The following figure supplement is available for figure 5:

**Figure supplement 1.** *fzo-1(tm1133)* suppresses extended lifespan of *raga-1(ok386)* without the use of FUDR.

DOI: https://doi.org/10.7554/eLife.49158.032

mitochondrial fragmentation. We generated animals carrying three perturbations: null mutation in *raga-1*, rescue of *raga-1* in neurons, and null mutation of *drp-1*, which we reasoned would block the ability of neuronal TORC1 to induce mitochondrial fragmentation in peripheral tissues. Notably, driving mitochondrial fusion by *drp-1* fully restores longevity in neuronal *raga-1* rescue animals to that of single *raga-1* mutants (*Figure 5f*). These data suggest that indeed, neuronal TORC1 modulates lifespan via cell nonautonomous effects on mitochondrial dynamics. In summary, our results highlight a critical role of neuronal TORC1 activity on healthy aging, and suggest that lifespan extension by reduced TORC1 signaling requires a fused mitochondrial network which itself can be modulated via TORC1 activity in neurons.

## Discussion

Taken together, our data suggest that AMPK and TORC1 can coordinate in neurons to modulate systemic organismal aging cell nonautonomously. Although primarily AMPK is regarded as an upstream suppressor of TORC1, our data suggest that for aging, lifespan extension via suppression of TORC1 activates and requires functional AMPK in neurons. However, lifespan extension by TORC1 suppression and AMPK activation in *C elegans* differ in two key aspects. First, expression of CRTC-1$^{S76A, S179A}$ in neurons completely suppresses AMPK longevity but has little effect on *raga-1* mutant lifespan. Second, whereas AMPK longevity requires both mitochondrial fusion and fission mechanisms to be functional (*Weir et al., 2017*), lifespan extension by *raga-1* deletion only requires mitochondrial fusion. Therefore, although both AMPK and TORC1 can modulate longevity and mitochondria dynamics cell nonautonomously in *C. elegans*, both the origin of those neuronal signals and the functional role mitochondria play in responding to them appear to differ.

Many of the described experiments used FUDR, a chemical inhibitor of thymidine synthesis, to block the production of progeny. While the actions of FUDR in postmitotic adult *C. elegans* are thought to act primarily in the germline to inhibit DNA replication in the dividing germ cells, there is also evidence that FUDR can act in somatic tissue and can lead to complex interactions with various genes and treatments in the regulation of lifespan (*Anderson et al., 2016*). As such, we chose a relatively small dose (40 μM) to use for our experiments, and importantly, verified key findings in lifespan experiments that did not use FUDR.

Beyond requiring LKB1/PAR-4, how TORC1 suppression activates AMPK in *C. elegans* remains unclear. S6K inhibits AMPK in mouse hypothalamus via S485 (or 491 depending on the isoform) (*Dagon et al., 2012*). Although our data suggest the conserved serine 551 residue is an inhibitory phosphorylation site in *C. elegans*, phosphorylation of AAK-2 S551 is not required for TORC1 mediated longevity. Several additional mechanisms have been recently identified that modulate AMPK activity, including an increasing number of post-translational modifications of the α subunit, expression levels of γ subunits and recruitment of LKB1 to lysosomal surface by the scaffold protein AXIN (*Hardie, 2014*; *Tullet et al., 2014*; *Zhang et al., 2016*). It remains to be tested whether these mechanisms are utilized by TORC1 to modulate AMPK in neurons, and whether they are relevant in the context of aging. Interestingly, *rsks-1* mutants require the creatine kinase, ARGK-1, which is primarily expressed in glial cells, to activate AMPK (*McQuary et al., 2016*). S6K1 has also been shown to inhibit AMPK in the hypothalamus in mice (*Dagon et al., 2012*). Together these data support a hypothesis that AMPK mediates TORC1 longevity in neurons.

While AMPK has been shown to act in neurons to promote longevity (*Burkewitz et al., 2015*; *Ulgherait et al., 2014*), one highlight of our study is that TORC1 itself has critical functions in neurons to modulate lifespan. It is especially striking that when TORC1 signaling is suppressed in all the major metabolically active tissues in *C. elegans* but active in neurons, animals are not long lived. *raga-1* or *rsks-1* neuronal rescued animals share non-aging related phenotypes with *raga-1* or *rsks-1* mutants, such as smaller body size, delayed development and reduced brood size (data not shown). These results suggest that neuronal TORC1 might modulate lifespan through a specific mechanism that is uncoupled from the broad effects of TORC1 on growth and anabolism, for example via its regulation of insulin-like or other neuropeptides that act systemically to regulate longevity. Understanding where and how RAGA-1 acts in the nervous system and whether suppressing TORC1 signaling only in neurons either genetically or pharmacologically is sufficient to promote healthy aging is now a key future goal.

Our data suggest that neuropeptide signals may act downstream of neuronal TORC1 to communicate with peripheral tissues. Indeed, we identified two insulin-like peptides, INS-6 and DAF-28, that are regulated by neuronal expression of *raga-1*. Both peptides have been shown to be regulated by food cues, and interestingly, when overexpressed are sufficient to suppress the long lifespan caused by loss of sensory signaling through the TAX-2/TAX-4 cyclic nucleotide gated channel (*Artan et al., 2016*). Whether these or other neuropeptides mediate the effects of neuronal TORC1 signaling on lifespan remains to be explored. With recent findings that neuropeptides influence aging in mammals (*Riera et al., 2014*), our study provides a critical starting point to investigate the identity and regulation of the neuropeptides and/or neurotransmitters that directly modulate TORC1 longevity.

Rapamycin suppresses TORC1 and alleviates a plethora of age-related pathologies and functional decline in mice, slowing age-related degenerative or neoplastic changes in liver, endometrium, heart and bone marrow (*Chen et al., 2009*; *Flynn et al., 2013*; *Wilkinson et al., 2012*). Moreover, short term rapamycin administration has significant anti-aging effects (*Bitto et al., 2016*). In addition, metformin, an anti-diabetic drug which both activates AMPK and inhibits TORC1 via AMPK-dependent and -independent mechanisms (*Howell et al., 2017*; *Kalender et al., 2010*), extends lifespan in nematodes and mice (*Martin-Montalvo et al., 2013*; *Onken and Driscoll, 2010*). The critical site of action for the anti-aging effects of rapamycin and metformin remain unknown. Our findings raise the exciting possibility that their effects on life- and healthspan could be mediated through their regulation of TORC1 and AMPK in the nervous system.

Finally, we identified mitochondria as the downstream 'receivers' of TORC1 mediated neuronal signals important to directly influence lifespan. Mitochondria are major organelles for energy and intermediate metabolite production, with critical roles in the aging process (*López-Lluch, 2017*). Our finding that *raga-1* mutant animals have hyperfused mitochondria is consistent with the in vitro mitochondrial hyperfusion induced by starvation or mTORC1 inhibition, which potentially allows more efficient ATP production and prevents healthy mitochondria from being degraded by mitophagy (*Gomes et al., 2011*). We further show that driving mitochondrial fission blocks the longevity of *raga-1* mutants, and driving fusion de-represses longevity in the presence of neuronal RAGA-1. Our results therefore unravel two critical aspects regarding the role of mitochondrial fusion in TORC1-mediated longevity: first, mitochondrial fission in peripheral tissues can be driven in a cell nonautonomous manner by neuronal RAGA-1; second, mitochondrial fusion is causally linked to *raga-1* longevity.

These and other recent work highlight how mitochondrial fission and fusion influence aging in a context-dependent manner: several longevity interventions require fusion (*Chaudhari and Kipreos, 2017*), while promoting fission can also extend lifespan in *Drosophila* (*Rana et al., 2017*), and dietary restriction and AMPK mediated longevity in *C. elegans* require both fusion and fission (*Weir et al., 2017*). Comparative analysis of the functional roles different mitochondrial network states play in AMPK and TORC1 longevity provide a unique opportunity to dissect out how mitochondrial networks might be modulated to promote healthy aging. Together, our data emphasize the role of TORC1 in the nervous system in modulating whole body metabolism and longevity. Further studies will help to elucidate the molecular identities of the neuronal signals and periphery receptors that underlie neuronal TORC1 activity to influence aging, and whether neuronal TORC1 might modulate aging in organisms beyond *C. elegans*.

## Materials and methods

**Key resources table**

| Reagent type (species) or resource | Designation | Source or reference | Identifiers | Additional information |
|---|---|---|---|---|
| Strain (*Caenorhabditis elegans*) | N2 | Caenorhabditis Genetics Center | WB Cat# N2_(ancestral), RRID:WB-STRAIN:N2_(ancestral) | Laboratory reference strain |
| Strain (*C. elegans*) | VC222 | Caenorhabditis Genetics Center | WB Cat# VC222, RRID: WB-STRAIN:VC222 | Genotype: *raga-1(ok386) II.* |

*Continued on next page*

*Continued*

| Reagent type (species) or resource | Designation | Source or reference | Identifiers | Additional information |
|---|---|---|---|---|
| Strain (*C. elegans*) | RB754 | Caenorhabditis Genetics Center | WB Cat# RB754, RRID: WB-STRAIN:RB754 | Genotype: *aak-2(ok524) X.* |
| Strain (*C. elegans*) | WBM997 | This study | | Genotype: *aak-2(wbm20) X.* |
| Strain (*C. elegans*) | RB1206 | Caenorhabditis Genetics Center | WB Cat# RB1206, RRID:WB-STRAIN :RB1206 | Genotype: *rsks-1(ok1255) III.* |
| Strain (*C. elegans*) | WBM536 | This study | | Genotype: *wbmEx238[rab-3p:: raga-1 cDNA:: SL2::mCherry::unc-54 3'UTR]* |
| Strain (*C. elegans*) | WBM772 | This study | | Genotype: *wbmEx333 [rab-3p::rsks-1 cDNA: :SL2::mCherry::unc-54 3'UTR]* |
| Strain (*C. elegans*) | WBM1167 | This study | | Genotype: *wbmIs79[eft-3p::3XFLAG::raga-1::SL2::wrmScarlet::unc-54 3'UTR, *wbmIs67]* |
| Strain (*C. elegans*) | WBM1168 | This study | | Genotype: *wbmIs80[rab-3p::3XFLAG:: raga-1::SL2::wrmScarlet:: rab-3 3'UTR, *wbmIs68]* |
| Strain (*C. elegans*) | WBM650 | This study | | Genotype: *wbmEx271 [ges-1p::raga-1 cDNA::SL2::mCherry::unc-54 3'UTR; rol-6 (su1006)]* |
| Strain (*C. elegans*) | WBM671 | PMID:29107506 | | Genotype: *wbmEx289 [myo-3p::tomm 20 aa1-49::GFP::unc54 3'UTR]* |
| Strain (*C. elegans*) | WBM955 | This study | | Genotype: *wbmEx373 [rab-3p::tomm-20 aa1-49::GFP::unc-54 3'UTR, rol-6]* |
| Strain (*C. elegans*) | WBM926 | PMID:29107506 | | Genotype: *wbmEx367[ges-1p::tomm20 aa1-49::GFP::unc-54 3'UTR]* |
| Strain (*C. elegans*) | CU5991 | Caenorhabditis Genetics Center | WB Cat# CU5991, RRID: WB-STRAIN:CU5991 | Genotype: *fzo-1 (tm1133) II.* |
| Strain (*C. elegans*) | CU6372 | Caenorhabditis Genetics Center | WB Cat# CU6372, RRID: WB-STRAIN:CU6372 | Genotype: *drp-1(tm1108) IV.* |
| Strain (*C. elegans*) | WBM861 | This study | | Genotype: *fzo-1(tm1133) II; wbmEx335 [rab-3p:3xFLAG fzo-1 cDNA: unc54 3'UTR, myo-3p:mCherry]* |
| Strain (*C. elegans*) | WBM612 | PMID:29107506 | | Genotype: *fzo-1 (tm1133) II; wbmEx258 [pHW11 (myo-3p:: 3xFLAG::fzo-1 cDNA::unc-54 3'UTR) + pRF4 (rol-6(SU1006))]* |
| Strain (*C. elegans*) | WBM639 | PMID:29107506 | | Genotype: *fzo-1(tm1133) II; wbmEx276 [pHW18 (ges-1p::3xFLAG::fzo-1 cDNA::unc54 3'UTR) + pRF4(rol-6(SU1006))]* |
| Antibody | Phospho-AMPKα (Thr172) antibody | Cell Signaling Technology | Cat# 2535, RRID:AB_331250 | |

*Continued on next page*

*Continued*

| Reagent type (species) or resource | Designation | Source or reference | Identifiers | Additional information |
|---|---|---|---|---|
| Antibody | Beta actin antibody | Abcam | Cat# ab8226, RRID:AB_306371 | |
| Software | MitoMAPR | This study | | Source code provided as *Source code 1* |
| Commercial assay or kit | TruSeq Stranded mRNA LT - Set A kit | Illumina | RS-122–2101 | |

## Worm strains

Worms were grown at 20°C on nematode growth media (NGM) plates seeded with *E. coli* strain OP50-1(CGC) with standard techniques (*Brenner, 1974*). Information for all strains used is in *Supplementary file 4*.

## RNAi feeding

Feeding RNAi clones were obtained from the Ahringer or Vidal RNAi libraries and sequence-verified before using. To use, bacteria were grown overnight in LB broth with 100 µg/mL carbenicillin and 12.5 µg/mL tetracycline, seeded on NGM plates with 100 µg/mL carbenicillin (NG Carb) and allowed 48 hr to grow at room temperature. At least 4 hr before use, 0.1M IPTG solution with 100 µg/mL carbenicillin and 12.5 µg/mL tetracycline was added to the bacterial lawn to induce dsRNA expression.

## Lifespan experiments

All worms were kept fed for at least two generations on OP50-1 bacteria. Before the start of each lifespan experiment, gravid adult worms were bleached and eggs were fed HT115 bacteria until adulthood to either start the lifespan experiment or a timed egg lay to obtain synchronized populations. Day 1 of lifespan marks the onset of egg laying. In the cases where FUDR is used, plates were seeded with bacteria, allowed 24 hr to grow and 100 µl of 1 mg/mL FUDR solution was seeded on top of the bacteria lawn for each plate containing 10 mL NGM (for a final concentration of 40 µM). FUDR was allowed 24 hr to diffuse to the whole plate before plates were used. When combining FUDR with RNAi treatments, to overcome potential inhibition of FUDR on dsRNA expression in bacteria, plates were induced with IPTG solution 18 hr after seeding; FUDR was applied 24 hr after seeding; IPTG was applied again 4 hr before use.

## Western blots

More than 500 day one adults were used for each sample. Worms were collected in M9 buffer with 0.01% Tween-20 and washed three times in M9. Liquids were removed after centrifugation and samples were frozen in liquid nitrogen. For worm lysis, RIPA buffer containing protease inhibitors (Sigma, MI, USA #8340) and phosphatase inhibitors (Roche, Basel, Switzerland #4906845001) was added to each sample at the same volume as the worm pellet. Worms were lysed via sonication (Qsonica, CT, USA Q700). Protein concentration was measured using Pierce BCA protein assay kit (Thermo Fisher Scientific, MA, USA PI23227) following manufacturer's instructions. To denature the proteins, 5X RSB was added and samples were heated to 95°C for 5 min. Samples containing 20–30 µg protein were loaded to 10% Tris-Glycine gels (Thermo Fisher Scientific #XP00100) for SDS-PAGE. Proteins were transferred to PVDF membranes (Thermo Fisher Scientific, #LC2005) and blocked with 5% BSA in TBST. Primary antibodies and dilutions are: phosphor-AMPK (Cell signaling, MA, USA #2535) 1:1000; beta actin (Abcam, Cambridge, UK #8226) 1:1000. Antibody signals were developed using ECL Western Blotting Detection Reagent (GE Healthcare, IL, USA Catalog number: 95038–560) and bands were quantified with Gel Doc system (Bio Rad) and Image Lab software (Version 4.1).

## Genotyping of deletion alleles

Worms were individually lysed in single worm lysis buffer and lysates were used as templates for PCR reactions with a combination of 2–3 primers that will produce bands of different sizes for wild type and mutant alleles. Primers and PCR conditions for each deletion allele are listed in *Supplementary file 5*.

## Generation of transgenes

To generate transgenic animals expressing *raga-1* in neurons, *raga-1* cDNA was PCR amplified and cloned using standard techniques into a plasmid where a 3X FLAG tag was added to the N terminus and a *gpd-2* SL2 sequence with mCherry ORF was added between *raga-1* stop codon and *unc-54* 3'UTR. *rab-3* promoter was subsequently PCR amplified and inserted. To express wild type and mutated forms of *aak-2*, both 3 kb promoter region before the *aak-2* gene and the 6.7 kb coding region were amplified from N2 genomic DNA and cloned into a plasmid, where a 3X FLAG tag and *unc-54* 3'UTR were added to the C terminus. Serine-to-alanine mutation was generated using Quik-Change II XL Site-directed Mutagenesis Kit (Agilent Technologies, CA, USA 200522) following manufacturer's instructions. Transgenic strains were generated via microinjection. Detailed information on strains used is in *Supplementary file 4*.

## Generation of single copy *raga-1* transgene knock-in strains

Strains WBM1167 *N2, wbmIs79[eft-3p::3XFLAG::raga-1::SL2::wrmScarlet::unc-54 3'UTR, *wbmIs67]* and WBM1168 *N2, wbmIs80[rab-3p::3XFLAG::raga-1::SL2::wrmScarlet::rab-3 3'UTR, *wbmIs68]* were generated by CRISPR according to *Silva-García et al. (2019)*. Specifically, a homology repair template (HR) containing *raga-1::SL2* sequences was amplified from plasmid pYZ30 using primers

'5' TATAAAGATCATGACATCGATTACAAGGATGACGAcGAtAAGTCTT CAAAACGAAAAGTT' and '

5' AACGCATGAACTCCTTGATAACTGCCTCTCCCTTGCTGACC ATGATGCGTTGAAGCAGTT',

followed by a second round of amplification using '5' CCGGGATGGACTACAAAGACCA TGACGGTGATTATAAAGATCATGACATCGATTACAAGG and '5' CGTGTCCGTTCATGGATCCC TCCATGTGGACCTTGAAACGCATGAACTCCTTGATA' to extend the HR arms. A CRISPR mix containing the raga-1::SL2 HR template, and a 3x flag 3' crRNA [5' TTACAAGGATGACGATGACA 3'] was prepared according to *Paix et al. (2015)*; *Silva-García et al. (2019)* and injected into strains WBM1143 and WBM1144 (*Silva-García et al., 2019*). Resulting CRISPR edited alleles, *wbmIs79* and *wbmIs80*, were outcrossed 6 and 5 times to N2 respectively to generate strains WBM1167 and WBM1168. To introduce *wbmIs79* and *wbmIs80* single copy transgenes into a *raga-1* mutant background the strains were crossed into WBM499 (an outcrossed *raga-1(ok386)* allele) and genotyped for the presence of the transgene by the visible expression of *wrmScarlet*, and for presence of the *ok386* deletion by PCR, using primers 5' TTCAAGTCCGAAACAGTCAATTCTC and 5' GGAAC TGAAGCGATCACACCGAC. *raga-1* rescue strains are WBM1169 *raga-1 (ok386) II; wbmIs79[eft-3p::3XFLAG::raga-1::SL2::wrmScarlet::unc-54 3'UTR, *wbmIs67]* and WBM1170 *raga-1 (ok386) II; wbmIs80[rab-3p::3XFLAG::raga-1::SL2::wrmScarlet::rab-3 3'UTR, *wbmIs68]*.

## Determination of developmental rate

Strains were fed and passaged for multiple generations on NG Carb plates seeded with HT1115 bacteria. At the start of the experiment, adult hermaphrodites were allowed to lay eggs for ~2 hr, after which, 50 eggs from each strain were picked to each of 2 new plates (100 eggs total) and left to develop at 20°C for 72–73 hr. The developmental stage was scored by eye under a light microscope on the basis of larval and adult stage specific hallmarks.

## Comparison of body size

Strains were fed and passaged for multiple generations on NG Carb plates seeded with HT1115 bacteria. To synchronize animals, L4-staged larvae were picked to a new plate and aged for two days at 20°C. Day two adult animals were picked to a new plate and anaesthetized in a drop of 0.4 mg/mL tetramisole for ~15 mins. Once the animals stopped moving, they were aligned into groups according to genotype, and imaged with an Axiocam camera on a Zeiss Discovery V8 dissection microscope.

## Microscopy

Worms of desired stage/age were anesthetized in 0.5 mg/mL tetramisole (Sigma, T1512) diluted in M9 and mounted to 2% agarose pads. For imaging of the mitochondrial TOMM-20 reporter in the muscle, images were taken using a Zeiss Imager.M2 microscope. Apotome optical sectioning was used to acquire fluorescence and one picture with best focus was chosen for each worm for quantification (as described in *Weir et al., 2017*). For imaging of mitochondria in neurons and in intestine, images were taken in the Sabri Ulker imaging lab using a Yokogawa CSU-X1 spinning disk confocal system (Andor Technology, South Windsor, CT, USA) combined with a Nikon Ti-E inverted microscope (Nikon Instruments, Melville, NY, USA). Images were taken using a 100x/1.45 oil Plan Apo objective lens, Zyla cMOS (Zyla 4.2 Plus USB3) camera and 488 nm Laser for GFP. Optical slice thickness was 0.2 µm. NIS elements software was used for acquisition parameters, shutters, filter positions and focus control. For images shown of the intestine, images were taken as a z stack and each plane was then threaded together by concatenation of the stack. These concatenated stacks were then rendered into 3d by the 3d viewer function of FIJI.

## *C. elegans* mitochondrial analysis using MitoMAPR

The images were analysed using a novel macro called MitoMAPR (*Source code 1*) in ImageJ or FIJI (*Schindelin et al., 2012*). Briefly, worm muscle cells and neurons were selected as ROIs, processed and filtered using the CLAHE plugin (*Zuiderveld, 1994*) with median filter and unsharp mask to increase the local contrast and particle distinctiveness. The ROI is then converted to a binary image to generate a 2D skeleton using the Skeletonize3D plugin (*Lee et al., 1994*). This skeleton image is then dissected using the AnalyzeSkeleton function (*Arganda-Carreras et al., 2010*) in FIJI to generate Tagged and Labelled skeletons. Labelled skeletons allow the program to count distinct mitochondrial objects while the tagged skeleton provides information about junction points, branch counts and other pre-configured attributes. MitoMAPR uses the values obtained from the Analyze-Skeleton function to quantify previously defined aspects of the mitochondrial network (*Koopman et al., 2006*). The workflow of the macro is illustrated in *Figure 4—figure supplement 1*. Analysis was performed in batch mode using the MitoMAPR_Batch (*Source code 1*, part B) while keeping all the parameters constant. While the user is required to select a region of interest (ROI) in case of images processed singly, MitoMAPR_Batch imports the saved ROIs generated while cropping the images for cells. A separate macro called CropR (Source code file 1, part C) was written to select and crop large data sets in batch mode. The cropped dataset is then used as a batch input for MitoMAPR_Batch. The codes for the macros can be found as supplementary notes.

The attributes used here to describe alterations in the mitochondrial architecture are listed in *Supplementary file 7*. Additionally, to determine the complexity of the mitochondrial network, we focused on the Network and Junction Point attribute. As illustrated in *Figure 4—figure supplement 2*, greater number of junction points in individual mitochondrial networks point towards higher complexity. The output data is kept as an array in a. CSV file that lists the values of all the above-mentioned attributes.

## RNA seq sample collection, RNA extraction and library preparation

Worms were bleached to HT115 bacteria carrying L4440 vector. Synchronized populations were obtained via timed egg lay. four biological replicate samples were collected for each genotype. For each sample, an independent egg lay was performed and 1000 mid-L4 stage progeny were harvested. For the *raga-1, rab-3p::raga-1::SL2::mCherry* strain, only worms carrying the transgene were picked out to a new NGM plate using a fluorescent dissecting microscope for subsequent collection. For N2 and *raga-1* mutant animals, worms were directly washed off the plates. M9 buffer with 0.01% Tween-20 were used to wash worms off the plates. Samples were centrifuged at 2,500 rpm to pellet the worms and washed three times with M9 buffer to remove bacteria. QIAzol lysis reagent (Qiagen, Venlo, Netherlands, #79306) was added to each sample before snap-freezing in liquid nitrogen. All samples were stored in −80°C freezer until RNA extraction.

To break the worm cuticle and improve RNA yield, all samples underwent five freeze-thaw cycles. In each cycle, samples were thawed at 37 °C and then snap-frozen in liquid nitrogen. RNA extraction was performed immediately using QIAGEN RNeasy Mini Kit (QIAGEN, #74104) following manufacturer's instructions. RNA quality was confirmed using Agilent Bioanalyzer 2100 (Agilent

Technologies). All samples passed the quality control standard of RIN > 8.0. mRNA libraries were prepared using TruSeq Stranded mRNA LT - Set A kit (Illumina, CA, USA RS-122–2101) following manufacturer's instructions and linked to different adaptors to enable pooling. Library quality was checked using 2200 High Sensitivity D1000 Tape Station (Agilent Technologies). Libraries were pooled and sequenced with Illumina HiSeq 2500 using 50-cycle, pair-end settings.

## RNA seq gene expression and functional analysis
### Read processing and quantification
All samples were processed using an RNA-seq pipeline implemented in the bcbio-nextgen project (https://bcbio-nextgen.readthedocs.org/en/latest/). Raw reads were examined for quality issues using FastQC (http://www.bioinformatics.babraham.ac.uk/projects/fastqc/) to ensure library generation and sequencing data were suitable for further analysis. Reads were aligned to the Ensembl94 build of the *C. elegans* genome using STAR (*Dobin et al., 2013*). Quality of alignments was assessed by checking for evenness of coverage, rRNA content, genomic context of alignments, complexity and other quality checks. Expression quantification was performed with Salmon (*Patro et al., 2017*) to identify transcript-level abundance estimates and then collapsed down to the gene-level using the R Bioconductor package tximport (*Soneson et al., 2015*).

Principal components analysis (PCA) and hierarchical clustering methods validated clustering of samples from the same sample group. Differential expression was performed at the gene level using the R Bioconductor package DESeq2 (*Love et al., 2014*). Differentially expressed genes were identified using the Likelihood Ratio Test (LRT) and significant genes were obtained using an FDR threshold of 0.01. Significant genes were separated into clusters based on similar expression profiles across the defined sample groups. Gene lists for each cluster was used as input to the R Bioconductor package clusterProfiler (*Yu et al., 2012*) to perform an over-representation analysis of Gene Ontology (GO) biological process terms. A secondary pairwise analysis was also performed for all pairs of sample groups using the Wald test.

## Quantitative real-time PCR
RNA was isolated from ~200 L4 staged (for validation of RNASeq in independent samples) or Day one adult (determination of *raga-1* and *ins-6* expression in adults) *C. elegans* samples using method described above. cDNA was synthesized from 30 μg of RNA with SuperScript VILO Master Mix (ThermoFisher Scientific, 11755050) following manufacturer's instructions. 5 ng cDNA was used as template for each RT-PCR reaction. 2 or three independent biological replicates were used for each genotype/condition and always run in parallel with Taqman control probe to invariant control gene Y45F10D.4 (Ce02467253_g1) (*Heintz et al., 2017*), for normalization on a 96 well plate. RT-PCR was performed on the StepOne Plus qPCR Machine (Life Technologies, MA, USA) using Taqman Universal Master Mix II (Life Technologies, 4440040). Taqman probes used to target each gene of interest are as follows: Ce02445578_g1 (C28A5.2), Ce02484227_g1 (F35E12.5 (*irg-5*)), Ce02421566_m1 (Y39G10AR.6 (*ugt-31*)), Ce02488119_g1 (K10G4.5), Ce02433249_g1 (ZK84.6 (*ins-6*)), Ce02489787_g1 (Y116F11B.1 (*daf-28*)) and Ce02439068_g1 (*raga-1*). Relative expression levels were calculated using ΔΔCt method.

## Statistics and reproducibility
Statistical methods are indicated in the figure legends. Statistical tests were performed using Graphpad Prism versions 7.0b and 8.0 for Mac. No animals or samples were excluded from the analyses. For lifespan analyses, *p* values were calculated using Log-rank (Mantel-Cox) test. Data from all lifespan experiments are included in *Supplementary file 6* without excluding any lifespan replicates. For qPCR, statistical significance was determined by two-tailed *t-test* unless otherwise indicated. Data are presented as mean ± s.e.m. P value: NS no significance, *<0.05, **<0.01, ***<0.001, ****<0.0001 relative to controls.

## Acknowledgements
We thank Caroline Heintz, Sneha Dutta, Kristopher Burkewitz, Carlos Silva-Garcia, Nicole Clark and Rohan Sehgal for assistance with lifespan assays, Arpit Sharma for help with imaging experiments,

Gunes Parlakgul for confocal expertise and training, and members of the Mair laboratory for scientific discussion and critical reading of the manuscript. We thank Nicola Neretti and Andrew Leith for assistance with initial data analysis.

## Additional information

### Funding

| Funder | Grant reference number | Author |
|---|---|---|
| National Institute on Aging | 1R01AG044346 | William, B Mair |
| National Institute on Aging | 1R01AG059595 | William Mair |

The funders had no role in study design, data collection and interpretation, or the decision to submit the work for publication.

### Author contributions

Yue Zhang, Conceptualization, Data curation, Formal analysis, Validation, Investigation, Visualization, Methodology, Writing—original draft, Writing—review and editing; Anne Lanjuin, Conceptualization, Data curation, Formal analysis, Validation, Investigation, Visualization, Writing—original draft, Project administration, Writing—review and editing; Suvagata Roy Chowdhury, Data curation, Software, Formal analysis, Validation, Visualization, Writing—original draft; Meeta Mistry, Data curation, Formal analysis; Carlos G Silva-García, Resources, Investigation; Heather J Weir, Conceptualization, Resources, Investigation, Visualization; Chia-Lin Lee, Emina Tabakovic, Validation; Caroline C Escoubas, Resources, Investigation, Visualization; William B Mair, Conceptualization, Supervision, Funding acquisition, Investigation, Writing—original draft, Project administration, Writing—review and editing

### Author ORCIDs

Anne Lanjuin (iD) https://orcid.org/0000-0003-2532-0863
William B Mair (iD) https://orcid.org/0000-0002-0661-1342

### Decision letter and Author response

Decision letter https://doi.org/10.7554/eLife.49158.045
Author response https://doi.org/10.7554/eLife.49158.046

## Additional files

### Supplementary files

• Source code 1. Code for muscle mitochondria analysis using MitoMAPR.
DOI: https://doi.org/10.7554/eLife.49158.033

• Supplementary file 1. DEGs from pairwise comparisons of RNA-Seq.
DOI: https://doi.org/10.7554/eLife.49158.034

• Supplementary file 2. Genes in each cluster from RNA-seq.
DOI: https://doi.org/10.7554/eLife.49158.035

• Supplementary file 3. GO terms enriched in each cluster.
DOI: https://doi.org/10.7554/eLife.49158.036

• Supplementary file 4. Worm strains.
DOI: https://doi.org/10.7554/eLife.49158.037

• Supplementary file 5. Genotyping strategies.
DOI: https://doi.org/10.7554/eLife.49158.038

• Supplementary file 6. Lifespan replicates.
DOI: https://doi.org/10.7554/eLife.49158.039

• Supplementary file 7. List of MitoMAPR attributes.
DOI: https://doi.org/10.7554/eLife.49158.040

• Transparent reporting form
DOI: https://doi.org/10.7554/eLife.49158.041

## Data availability

Sequencing data have been deposited in GEO under accession code GSE132794.

The following dataset was generated:

| Author(s) | Year | Dataset title | Dataset URL | Database and Identifier |
|---|---|---|---|---|
| Zhang Y, Mair W | 2019 | RNA seq of raga-1 neuronal rescue *C. elegans* | https://www.ncbi.nlm.nih.gov/geo/query/acc.cgi?acc=GSE132794 | NCBI Gene Expression Omnibus, GSE132794 |

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
