## [Decision Letter]

Thank you for submitting your work entitled "Neuronal TORC1 modulates longevity via AMPK and cell nonautonomous regulation of mitochondrial dynamics in *C. elegans*" for consideration by *eLife*. Your article has been reviewed by a Senior Editor, a Reviewing Editor, and three reviewers. The following individual involved in review of your submission has agreed to reveal their identity: Seung-Jae Lee (Reviewer #2).

Our decision has been reached after consultation between the reviewers. Based on these discussions and the individual reviews below, we regret to inform you that your work will not be considered further for publication in *eLife*.

The reviewers appreciated the importance of the work, but raised some substantial concerns regarding methodology and interpretation. In particular, the use of extrachromosomal arrays and different genetic backgrounds for various experiments raises potential concerns about technical artifacts. Due to the extensive and open-ended nature of the essential revisions, I felt that it would be difficult or impossible to address these concerns within the timeframe. If you are able to address these concerns, a suitably revised manuscript could be considered again at *eLife*.

Essential revisions:

1) Use of extrachromosomal arrays. Most experiments using transgenes were done with extrachromosomal arrays. Minimally, multiple extrachromosomal array lines should be used to verify the results. Extrachromosomal arrays are subject to random segregation between cells and rearrange the DNA fragments in ways that cause unnatural expression of the transgenes, particularly in gut cells; thus the results about gut expression of an extrachromosomal array are not really worth considering until one observes the same thing with an integrated reporter, preferably single copy at some defined, intergenic site in the genome- or a knock-in. Positive results with arrays are more believable than negative results, but both can be misleading. Also, worth considering, extrachromosomal arrays and multicopy insertions into random genomic loci can also cause lifespan-extending dye filling defects in neurons – case and point would be the overexpression sirtuin line that lived a long time, not because of sirtuin overexpression, but because of a dye-filling defect in the neurons.

2) Inconsistent use of a roller background in some experiments. roller genetic background with their extrachromosomal arrays. Roller animals perpetually twist and do not have the same behavior as wild-type animals. So, comparing animals that are rolling to animals that are not, using extrachromosomal arrays, is riddled with technical concerns.

3) Use of FUDR without controlling for potential effects of the drug. FUDR inhibits DNA synthesis, including synthesis of mitochondrial DNA. FUDR also increases the lifespan of at least one mitochondrial mutant (*gas-1*). Thus, there is reason to believe that FUDR interacts with mitochondria and that is sufficient to warrant the testing of these phenomena in the absence of FUDR to ensure what they are reporting is not FUDR dependent.

4) In Figure 4, the authors showed that mitochondrial fusion-defective *raga-1(-); fzo-1*(-) mutants did not live long, suggesting that *raga-1* mutations increase lifespan through elevating fused mitochondria. Please show the mitochondrial patterns of *rag-1(-), raga-1(-); fzo-1*(-), and *raga-1(-); drp-1*(-) mutants to strengthen their findings.

5) In Figure 4B and C, the authors showed that intestinal *fzo-1* is required for the lifespan extension by *raga-1* mutations. However, they did not test muscles where neuronal *raga-1* rescue restored the mitochondrial dynamics of *raga-1* mutants. They need to test whether muscle-specific *fzo-1* is also required for the lifespan extension by *raga-1* mutations and test whether neuronal *raga-1* rescue restored the mitochondrial dynamics changes by *raga-1* mutations in the intestine as well.

Reviewer #1:

The authors of this paper have done a great job writing clearly and analyzing the data to genetically investigate how neuronal *raga-1* loss (mTORC1 component) increases longevity. The results are generally interesting in terms of understanding cell nonautonomous regulation of organelles (mitochondria), and important for the field of aging.

However, my enthusiasm for the manuscript diminished when I discovered the questionable technical quality of the transgene work and use of FUDR without an FUDR-less control to show that the phenomena are not FUDR-dependent. I want this study to be published, and wish the authors great success, but as it stands, the manuscript needs additional work to be sure of the results. Alternatively, they could implement heavy, explicit, technical caveats warning readers of the uncertainties in their results. Given the confident writing style, I believe the authors would/should probably just make better transgenes and try the experiments without FUDR. In worms, they could get their strains in hand in three weeks, split them into different biological replicates and perform a few lifespan experiments in parallel, or in tight sequential fashion in a couple of months without FUDR.

Most experiments using transgenes were done with extrachromosomal arrays. Extrachromosomal arrays are randomly arranged concatemers of injected DNA that are inherited in a non-mendelian fashion. Positive results are only preliminary and, in the past, before single copy reporter gene technology was available with Mos transposon (2008) and CRISPR based (2012/2013) methods, researchers would integrate the arrays and/or examine multiple insert lines to verify their findings. Minimally, multiple extrachromosomal array lines should have been used to verify the results; we have been given no such data. Extrachromosomal arrays are subject to random segregation between cells and rearrange the DNA fragments in ways that cause unnatural expression of the transgenes, particularly in gut cells; thus the results about gut expression of an extrachromosomal array are not really worth considering until one observes the same thing with an integrated reporter, preferably single copy at some defined, intergenic site in the genome- or a knock-in. Positive results with arrays are more believable than negative results, but both can be misleading. And, in 2018, this dated technology is no longer the standard or sufficient; rather, it should be used as preliminary data. It’s also worth considering that extrachromosomal arrays and multicopy insertions into random genomic loci can cause lifespan-extending dye filling defects in neurons – case and point would be the overexpression sirtuin line that lived a long time, not because of sirtuin overexpression, but because of a dye-filling defect in the neurons. I don't want to see another group get burned because of poor transgene practices. I believe that this group is doing great work and has noble intent, but simply needs a bit of guidance.

Additionally, they inconsistently use a roller genetic background with their extrachromosomal arrays. Roller animals perpetually twist and do not have the same behavior as wild-type animals. So, comparing animals that are rolling to animals that are not, using extrachromosomal arrays, is riddled with technical concerns. That is a setup for a potential incorrect answer/failure.

FUDR inhibits DNA synthesis, including synthesis of mitochondrial DNA. FUDR also increases the lifespan of at least one mitochondrial mutant (*gas-1*). Thus, there is reason to believe that FUDR interacts with mitochondria and that is sufficient to warrant the testing of these phenomena in the absence of FUDR to ensure what they are reporting is not FUDR dependent.

Again, this is a well written paper, and the results may end up being correct, but there is sufficient technical concern to ask for more data before letting the authors publish something that has several potential technical artifacts that could easily be addressed with a few modern transgenes and experiments without FUDR. This is a fantastic lab that seems to be doing great things and I want to see the technical quality of their work reflect their apparent high intellectual capacity.

Reviewer #2:

Target of rapamycin (mTOR) and AMP-activated protein kinase (AMPK) are key mediators of longevity in various species. However, how mTOR and AMPK coordinate to promote longevity remains unclear. Here, Zhang et al. suggest that activation of AMPK and inhibition of TORC1 extend lifespan via rather distinct mechanisms. They further showed that neuronal TORC1 regulates lifespan by modulating mitochondrial dynamics in other tissues through activating neuronal AMPK. This is an excellent paper that addressed mechanisms of lifespan extension by TORC1 and AMPK pathways. The experiments were generally rigorously executed and the paper is well written. My specific comments are as follows.

1) Through the entire paper, the authors claim that AMPK and mTOR regulates lifespan through distinct mechanisms. However, most of the results explain how AMPK acted in mTOR pathways in detail; inhibition of neuronal mTOR pathway extends lifespan possibly through neuronal AMPK. In addition, the lines of supporting evidence for the distinct aging-regulatory mechanisms by CA-AAK and inhibition of mTOR pathway are indirect. Please tone down the claims.

2) In Figure 4, the authors showed that mitochondrial fusion-defective *raga-1(-); fzo-1*(-) mutants did not live long, suggesting that *raga-1* mutations increase lifespan through elevating fused mitochondria. Please showthe mitochondrial patterns of *rag-1(-), raga-1(-); fzo-1*(-), and *raga-1(-); drp-1*(-) mutants to strengthen their findings.

3) In Figure 4B and C, the authors showed that intestinal *fzo-1* is required for the lifespan extension by *raga-1* mutations. However, they did not test muscles where neuronal *raga-1* rescue restored the mitochondrial dynamics of *raga-1* mutants. They need to test whether muscle-specific *fzo-1* is also required for the lifespan extension by *raga-1* mutations and test whether neuronal *raga-1* rescue restored the mitochondrial dynamics changes by *raga-1* mutations in the intestine as well.

Reviewer #3:

In this manuscript, the authors seek to 1) establish a relationship between AMPK and neuronal TORC1 activity in the context of longevity, and 2) better understand the mechanism of TORC1 inhibition slowing aging in *C. elegans*. They find that the downstream longevity pathways of AMPK and TORC1 diverge and proceed to investigate the cell nonautonomous pathway initiated by neuronal TORC1 activity. They then investigate the downstream effects of the pathway by characterizing the mitochondrial networks of wildtype and mutants with suppressed neuronal TORC1 activity before associating the differences in the mitochondrial properties with the longevity of the latter animals.

In the opinion of this reviewer, the claims made by the authors are largely substantiated, including:

1) AMPK is required for longevity mediated by TORC1 suppression: Substantiated

2) TORC1 and AMPK act via separable downstream mechanisms: Substantiated.

3) Neuronal AMPK is required for TORC1-mediated longevity: Mostly substantiated.

4) TORC1 acts in neurons to regulate aging: Substantiated.

5) Neuronal RAGA-1 does not reverse known pro-longevity mechanisms induced by global TORC1 suppression: Substantiated.

6) Neuronal RAGA-1 modulates aging via neuropeptide signaling: Substantiated.

7) Neuronal RAGA-1 drives peripheral mitochondrial fragmentation in aging animals: Substantiated.

8) *raga-1* deletion specifically requires a fused mitochondrial network to promote longevity: Substantiated.

9) Neuronal RAGA-1 suppresses lifespan cell nonautonomously via mitochondrial fission: Substantiated.

The major question to this reviewer about this manuscript is not whether the data are substantiated but whether the advance is substantial enough for this journal. The authors establish a relationship between AMPK and TORC1 (which was already known to exist but further defined here), but don't really investigate it beyond cursory genetic and biochemical characterization. They then find a cell-specific role for TORC1 and suggest a signaling mechanism (neuropeptides), but do not identify any signals. Finally, they establish mitochondrial dynamics as being necessary, but offer little insight as to how fission and more importantly, fusion, might play a mechanistic role. In short, while the paper is geared towards investigating neuronal TORC1 and how it affects longevity, they don't really progress or analyze the relationship between the two based on their own data. It would be nice to see a couple of the "additional mechanisms" (mentioned in the second paragraph of the Discussion) they favor based off their work.

[Editors' note: further revisions were requested prior to acceptance, as described below.]

Thank you for submitting your article "Neuronal TORC1 modulates longevity via AMPK and cell nonautonomous regulation of mitochondrial dynamics in *C. elegans*" for consideration by *eLife*. Your article has been reviewed by and Anna Akhmanova as the Senior Editor, a Reviewing Editor, and three reviewers. The following individuals involved in review of your submission have agreed to reveal their identity: Seung-Jae V Lee (Reviewer #1); Scott F Leiser (Reviewer #2).

The reviewers have discussed the reviews with one another and the Reviewing Editor has drafted this decision to help you prepare a revised submission.

The reviewers all agreed that the authors have substantially improved the manuscript from the initial submission and that this will represent a significant contribution. There were some remaining concerns about clarity and interpretation, which I would ask the authors to addressed by modifying the text appropriately.

Please make sure genetic nomenclature is correct. Following are some examples of errors:

Figure 5A: *“raga-1, fzo-1*” should be “*fzo-1 raga-1*” without comma.

Figure 5B: “*raga-1; drp-1*” should be all italic, even the semicolon.

In some cases it is unclear which experiments were done with extrachromosomal arrays and which were done with new single copy insertions. Please ensure that this is clearly noted.

The extrachromosomal array prab-3 ex is not listed in the strain list. And the lifespan data does not say which strains were used for which experiments. Therefore, the supplements showing which strains were used seem incomplete and the other lifespan file does not detail which strain was used in which experiment. The lifespan data should say which strains were used – not just the genotype – and indicate if they are single copy insertions or extrachromosomal arrays.

Please provide the NA of the 100X objective used for imaging. It will be critical for anyone wanting to reproduce this. I'm guessing it's 1.4, but it would be better to know.

The microscopy in Figure 4 – the images of mitochondrial networks in intestines are of terrific quality. How are these images shown? Are they single image planes? Are they z projections? More detail here would again help people. Even though you may have it in another publication, many would appreciate a sentence or two describing the image plane used for the image cytometry in the methods.

One reviewer had significant concerns regarding the use of FUDR and the roller background in some experiments. Although these methods are still common in the field, it would perhaps be useful to address these concerns in the text, and if data are available that do not have these potential confounds those could be included as well. In particular, this reviewer noted that FUDR enhances the effects of mitochondrial double mutants (*fzo/drp*) from your prior work and wonders if the use of FUDR could be impacting outcomes in the current study.

---

## [Author Response]

Essential revisions:1) Use of extrachromosomal arrays. Most experiments using transgenes were done with extrachromosomal arrays. Minimally, multiple extrachromosomal array lines should be used to verify the results. Extrachromosomal arrays are subject to random segregation between cells and rearrange the DNA fragments in ways that cause unnatural expression of the transgenes, particularly in gut cells; thus the results about gut expression of an extrachromosomal array are not really worth considering until one observes the same thing with an integrated reporter, preferably single copy at some defined, intergenic site in the genome- or a knock-in. Positive results with arrays are more believable than negative results, but both can be misleading. Also, worth considering, extrachromosomal arrays and multicopy insertions into random genomic loci can also cause lifespan-extending dye filling defects in neurons – case and point would be the overexpression sirtuin line that lived a long time, not because of sirtuin overexpression, but because of a dye-filling defect in the neurons.We thank the reviewers for their comments and concerns here about the use of and variability of extrachromosomal arrays, and random mutagenesis integration methods. Current methods for single copy tissue specific rescue involve MosSCI insertion, which invariably requires co-insertion of a rescue marker (usually *unc-119*), which can impact neuronal signaling. Therefore, since no usable technique was available to generate the strains needed to alleviate the reviewer’s concerns, we decided to build one. Since the original submission we have made from scratch and published a CRISPR based strategy – “Single Copy Knock-in Loci for Defined Gene Expression (SKI LODGE) in *C. elegans*” (Silva-Garcia, 2019). Building this system required CRISPR insertion and validation at safe harbor loci in intergenic regions of entire promoter sequences, upstream of a landing pad crRNA target sequence. SKI LODGE strains therefore allow insertion of a cDNA of choice at a known locus under the regulation of an ectopic tissue specific promoter. We are proud of this new system, built in response to the concerns of the reviewers, and have already sent all the strains to many labs on request and the CGC. We therefore hope that this concerted effort and resource will not only allow our data and methodologies to improve now and in the future, but support the whole field. For the purposes of this revision, we then used the pan-neuronal SKI LODGE strain to validate the key point of the paper, namely the neuronal effects of RAGA-1 on longevity. The single copy SKI LODGE neuronal RAGA-1 rescue line expresses stably and only in neurons (Silva-Garcia, 2019). We compared lifespan, growth and development in *raga-1* mutants with single copy SKI LODGE generated rescue in neurons or all somatic tissue (New Figure 2). Critically and supporting the original data, single copy rescue of RAGA-1 in neurons suppressed *raga-1* mutant longevity. These lines have no roller insertion. Expression of *raga-1* in the single copy neuronal rescue was very low compared to the somatic SKI LODGE line or the previous neuronal Ex line (Figure 2E). Strikingly, although single copy neuronal RAGA-1 rescue restored WT longevity, it did not affect body size (Figure 2F) – these data suggest that the effects of TORC1 on growth and lifespan can be uncouple via modulation in discreet tissues.2) Inconsistent use of a roller background in some experiments. roller genetic background with their extrachromosomal arrays. Roller animals perpetually twist and do not have the same behavior as wild-type animals. So, comparing animals that are rolling to animals that are not, using extrachromosomal arrays, is riddled with technical concerns.

We agree with the reviewer and hope the SKI LODGE system will alleviate the need for any use of Ex line with roller in the future. We have used SKI LODGE to replicate the key finding of the paper (see above). However, we believe it was beyond the scope of feasibility for us to re-do the entire body of work using SKI LODGE: the original submission was a PhD thesis performed over 6 years, much of it before CRISPR was widely available. However, in the small number of experiments that have used *rol-6* in the manuscript, the key comparison statistically is always between two roller lines. Therefore, we do not believe behavioral differences account for the effects we see.

3) Use of FUDR without controlling for potential effects of the drug. FUDR inhibits DNA synthesis, including synthesis of mitochondrial DNA. FUDR also increases the lifespan of at least one mitochondrial mutant (gas^-1^). Thus, there is reason to believe that FUDR interacts with mitochondria and that is sufficient to warrant the testing of these phenomena in the absence of FUDR to ensure what they are reporting is not FUDR dependent.

We thank the reviewers for this point and have added new data replicating the effects we see of neuronal RAGA-1 suppression of longevity and *fzo-1* suppression of *raga-1* longevity without FUDR (Figure 2—figure supplement 1 and Figure 5—figure supplement 1). All lifespan experiments done during revision did not use FUDR, including those with the SKI LODGE lines.

4) In Figure 4, the authors showed that mitochondrial fusion-defective raga-1(-); fzo-1(-) mutants did not live long, suggesting that raga-1 mutations increase lifespan through elevating fused mitochondria. Please show the mitochondrial patterns of rag-1(-), raga-1(-); fzo-1(-), and raga-1(-); drp-1(-) mutants to strengthen their findings.

We thank the reviewers for this point and have added new data replicating the effects we see of neuronal RAGA-1 suppression of longevity and *fzo-1* suppression of *raga-1* longevity without FUDR (Figure 2—figure supplement 1 and Figure 5—figure supplement 1). All lifespan experiments done during revision did not use FUDR, including those with the SKI LODGE lines.

5) In Figure 4B and C, the authors showed that intestinal fzo-1 is required for the lifespan extension by raga-1 mutations. However, they did not test muscles where neuronal raga-1 rescue restored the mitochondrial dynamics of raga-1 mutants. They need to test whether muscle-specific fzo-1 is also required for the lifespan extension by raga-1 mutations and test whether neuronal raga-1 rescue restored the mitochondrial dynamics changes by raga-1 mutations in the intestine as well.

We thank the reviewer for this point. We have added new data showing that muscle FZO-1 is not sufficient to rescue *raga-1* mutant longevity (Figure 5D). We have also added new data showing that *raga-1* deletion maintains mitochondrial fusion with age in the intestine Figure 4—figure supplement 4).

[Editors' note: further revisions were requested prior to acceptance, as described below.]

The reviewers all agreed that the authors have substantially improved the manuscript from the initial submission and that this will represent a significant contribution. There were some remaining concerns about clarity and interpretation, which I would ask the authors to addressed by modifying the text appropriately.Please make sure genetic nomenclature is correct. Following are some examples of errors:Figure 5A: “raga-1, fzo-1” should be “fzo-1 raga-1” without comma.Figure 5B: “raga-1; drp-1” should be all italic, even the semicolon.

Thank you for bringing this to our attention. We have corrected the nomenclature in Figure 5 and in the supporting files describing the strains used for each experiment and their genotypes. In those cases where the genotypes have been abbreviated for simplicity in the figure, we have directed the reader to the full genotype information provided in the Supporting files (Supplementary file 4, Supplementary file 6).

In some cases it is unclear which experiments were done with extrachromosomal arrays and which were done with new single copy insertions. Please ensure that this is clearly noted.

Thank you, we have clarified where necessary in the text and have amended the figures to show more explicitly which experiments were done using extrachromosomal arrays and which were done by single copy knock-in. In addition, we have added a column describing the nature of the transgene used in each strain (extrachromosomal vs. integrated multi copy array vs. single copy knock in) in Supplementary file 4 and in Supplementary file 6.

The extrachromosomal array prab-3 ex is not listed in the strain list. And the lifespan data does not say which strains were used for which experiments. Therefore, the supplements showing which strains were used seem incomplete and the other lifespan file does not detail which strain was used in which experiment. The lifespan data should say which strains were used – not just the genotype – and indicate if they are single copy insertions or extrachromosomal arrays.

We apologize for the insufficient strain information initially provided in the supplements. We now provide strain numbers for all strains used and note whether the transgene used in each strain is extrachromosomal, an integrated multicopy array, or present as single cope insertion.

Please provide the NA of the 100X objective used for imaging. It will be critical for anyone wanting to reproduce this. I'm guessing it's 1.4, but it would be better to know.

Thank you, the 100x objective we used was an NA of 1.45. We have noted that information in the subsection “Microscopy”.

The microscopy in Figure 4 – the images of mitochondrial networks in intestines are of terrific quality. How are these images shown? Are they single image planes? Are they z projections? More detail here would again help people. Even though you may have it in another publication, many would appreciate a sentence or two describing the image plane used for the image cytometry in the methods.

We thank the reviewers for bringing up this question. We have included the following additional information on how we acquired and processed images of mitochondria in the intestine and in the Materials and methods section:

“For images shown of the intestine, images were taken as a z stack and each plane was then threaded together by concatenation of the stack. These concatenated stacks were then rendered into 3d by the 3d viewer function of FIJI.”

One reviewer had significant concerns regarding the use of FUDR and the roller background in some experiments. Although these methods are still common in the field, it would perhaps be useful to address these concerns in the text, and if data are available that do not have these potential confounds those could be included as well.

We thank the reviewer for the suggestion and acknowledge that the practice of using FUDR as part of lifespan and other experiments is not ideal. We have amended the text further to address this concern, and in addition added the following to the Discussion section:

“Many of the described experiments used FUDR, a chemical inhibitor of thymidine synthesis, to block the production of progeny. While the actions of FUDR in postmitotic adult *C. elegans* are thought to act primarily in the germline to inhibit DNA replication in the dividing germ cells, there is also evidence that FUDR can act in somatic tissue and can lead to complex interactions with various genes and treatments in the regulation of lifespan (Anderson et al., 2016). As such, we chose a relatively small dose (40 mM) to use for our experiments, and importantly, verified key findings in lifespan experiments that did not use FUDR.”

In particular, this reviewer noted that FUDR enhances the effects of mitochondrial double mutants (fzo/drp) from your prior work and wonders if the use of FUDR could be impacting outcomes in the current study.

We are unclear what data the reviewer is referring to here. Indeed, in the paper cited (Weir et al., 2017), we performed lifespans of *fzo-1* mutants with and without FUDR and saw little difference between with FUDR and without. As seen previously by us and others, lifespans on HT115 with carb are longer lived than those on OP50 without antibiotics. Therefore, we do not know the result being referred to by the reviewer ‘that FUDR enhances the effects of the *fzo-1; drp-1*

double mutant’.

However, we also note that we have replicated the key FZO-1 experiment in this study demonstrating that the mitochondrial fusion mutant *fzo-1* blocks *raga-1* lifespan both with and without using FUDR, suggesting it is not an artifact or some complex interaction caused by FUDR use (compare Figure 5A, Figure 5—figure supplement 1). In addition, all of the experiments we conducted using the single copy knock-in strains (lifespans, body size determination, developmental delay) were done without the use FUDR.